SciPost Physics     

# To Profile or To Marginalize – A SMEFT Case Study

Ilaria Brivio[1,2], Sebastian Bruggisser[1,3], Nina Elmer[1], Emma Geoffray[1], Michel Luchmann[1],
and Tilman Plehn[1]

**1** Institut für Theoretische Physik, Universität Heidelberg, Germany
**2** Physik Institut, University of Zurich, Switzerland
**3** Department of Physics and Astronomy, Uppsala University, Sweden

December 13, 2022

## Abstract

Global SMEFT analyses have become a key interpretation framework for LHC physics, quantifying how well a large set of kinematic measurements agrees with the Standard Model. This agreement is encoded in measured Wilson coefficients and their uncertainties. A technical challenge of global analyses are correlations. We compare, for the first time, results from a profile likelihood and a Bayesian marginalization for a given data set with a comprehensive uncertainty treatment. Using the validated Bayesian framework we analyse a series of new kinematic measurements. For the updated dataset we find and explain differences between the marginalization and profile likelihood treatments.

# 1 Introduction

Higgs physics at the LHC [1] perfectly illustrates a deep tension in contemporary particle physics: on the one hand, the existence of a fundamental Higgs boson is a direct consequence of describing the electroweak gauge sector in terms of a quantum field theory, specifically a renormalizable gauge theory. It looks like Nature chose the simplest possible realization of the Higgs mechanism, with one light scalar particle and an electroweak vacuum expectation value (VEV) of unknown origin. On the other hand, puzzles like dark matter or baryogenesis seem to point to non-minimal Higgs sectors for convincing solutions based on renormalizable quantum field theory, but without any LHC hint in these directions. The main goal of the LHC Higgs program is to understand if Nature really took the opportunity of a minimal electroweak and Higgs sector solving as many problems as possible, or why she skipped this opportunity in favor of theoretically less attractive alternatives. Or, more practically speaking, we need to study as many Higgs properties as precisely as possible.

Given the vast LHC dataset already after Run 2 and our fundamental ignorance of the correct UV-completion of the Standard Model (SM), we need to measure Higgs-related observables and express them in a consistent, fundamental, and comprehensive theory framework. To provide the necessary precision, this framework has to be defined beyond leading order in perturbation theory, it needs to incorporate kinematic information, and it should allow us to combine as many LHC observables as possible. The EFT extension of the Standard Model (SMEFT) [2] fulfills precisely these three requirements and defines a theoretical path to understanding the entire LHC dataset in terms of a fundamental Lagrangian. Its main shortcoming is the necessary truncation in the operator dimensionality. The truncated SMEFT approximation will hardly describe new physics appropriately, so SMEFT should really be viewed as a systematic, conservative limit-setting tool. Of course, this practical aspect does not cut into the fundamental attractiveness of an effective quantum field theory description of all LHC data.

While ATLAS and CMS have not published properly global SMEFT analyses of the Higgs-electroweak or top sector, there exists a range of phenomenological Higgs-gauge analyses [3–8], top analyses [9–11], combinations of the two [12,13], and combinations with parton densities [14]. These analyses are typically based on experimentally preprocessed information, including the full range of uncertainties. Given that by assumption any SMEFT analysis will be centered around the renormalizable SM-Lagrangian, the main focus of all global analyses is the uncertainty treatment and the correlations between the different operators. Technically, these two tasks tend to collide. We can choose a conservative uncertainty treatment based on profile likelihoods and nuisance parameters, but it is much more computing-efficient to treat correlations through covariance matrices of marginalized Gaussian likelihoods [15]. The SFitter framework [3,11,16–19] is unique in the sense that it has mostly been used for profile likelihood analyses, but can provide marginalized limits equally well [16,20].

We make use of this flexibility and study, for the first time, the difference between profiled and marginalized likelihoods of the same global Run 2 dataset. In Sec. 3 we find that for the Higgs-electroweak dimension-6 operators and the given dataset the two approaches agree well, so we can use the marginalized setup to treat correlated measurements and uncertainties efficiently. Based on these results, we include a range of recent Run 2 measurements from Higgs studies as well as from exotics resonance searches, again with a focus on a comprehensive and conservative uncertainty treatment, in Sec. 4. Finally, we study the impact of these new measurements and the inputs from a global top analysis in Sec. 5 and find interesting differences between the profiling and marginalization methods.

## 2   SMEFT Lagrangian

The SMEFT Lagrangian is based on the same field content, global, and gauge symmetries as the SM. It includes interactions with canonical dimension larger than four, organised in a systematic expansion in the inverse powers of a new physics scale [21–23]. Neglecting lepton number violation at dimension five, the leading beyond-SM effects stem from dimension-six terms,

$$\mathcal{L} = \sum_x \frac{f_x}{\Lambda^2}\, \mathcal{O}_x \;. \tag{1}$$

There are 59 baryon-number conserving operators, barring flavor structure [24–28]. We use the operator basis of Refs. [3,29], starting with a set of $P$-even and $C$-even operators and then using the equations of motion to define a basis without blind directions in the electroweak precision data. We neglect operators that cannot be studied at the LHC yet, like those changing the triple-Higgs vertex [30–34]. We also neglect operators which are too strongly constrained from other LHC measurements to affect the Higgs-electroweak analysis, like the ubiquitous triple-gluon operator

$$\mathcal{O}_G = f_{ABC} G_{A\nu}^{\rho} G_{B\lambda}^{\nu} G_{C\rho}^{\lambda}\,, \tag{2}$$

which is strongly constrained from multi-jet production [35]. In the bosonic sector the relevant operators then are

$$\begin{aligned}
&\mathcal{O}_{GG} = \phi^\dagger \phi\, G_{\mu\nu}^a G^{a\mu\nu} &&\mathcal{O}_{WW} = \phi^\dagger\, \hat{W}_{\mu\nu}\hat{W}^{\mu\nu}\, \phi &&\mathcal{O}_{BB} = \phi^\dagger\, \hat{B}_{\mu\nu}\hat{B}^{\mu\nu}\, \phi \\
&\mathcal{O}_{W} = (D_\mu\phi)^\dagger \hat{W}^{\mu\nu}(D_\nu\phi) &&\mathcal{O}_{B} = (D_\mu\phi)^\dagger \hat{B}^{\mu\nu}(D_\nu\phi) &&\mathcal{O}_{BW} = \phi^\dagger\, \hat{B}_{\mu\nu}\hat{W}^{\mu\nu}\, \phi \\
&\mathcal{O}_{\phi 1} = (D_\mu\phi)^\dagger\, \phi\phi^\dagger\, (D^\mu\phi) &&\mathcal{O}_{\phi 2} = \frac{1}{2}\partial^\mu(\phi^\dagger\phi)\partial_\mu(\phi^\dagger\phi) \\
&\mathcal{O}_{3W} = \mathrm{Tr}\left(\hat{W}_{\mu\nu}\hat{W}^{\nu\rho}\hat{W}_{\rho}^{\mu}\right)\,,
\end{aligned} \tag{3}$$

where $\hat{B}_{\mu\nu} = ig' B_{\mu\nu}/2$ and $\hat{W}_{\mu\nu} = ig\sigma^a W_{\mu\nu}^a/2$. The covariant derivative acting on the Higgs doublet is $D_\mu = \partial_\mu + ig' B_\mu/2 + ig\sigma_a W_\mu^a/2$.

In addition to the purely bosonic operators, we also need to include single-current operators modifying the Yukawa couplings,

$$\begin{aligned}
\mathcal{O}_{e\phi,22} &= \phi^\dagger\phi\, \bar{L}_2\phi e_{R,2} &\qquad \mathcal{O}_{e\phi,33} &= \phi^\dagger\phi\, \bar{L}_3\phi e_{R,3} \\
\mathcal{O}_{u\phi,33} &= \phi^\dagger\phi\, \bar{Q}_3\tilde{\phi}u_{R,3} &\qquad \mathcal{O}_{d\phi,33} &= \phi^\dagger\phi\, \bar{Q}_3\phi d_{R,3}\,,
\end{aligned} \tag{4}$$

The main difference to earlier SFitter analyses is that we treat the correction to the muon Yukawa $f_\mu$ as an independent parameter, while previously it was tied to $f_\tau$ via an approximate flavor symmetry. As LHC Run 2 found experimental evidence for the Higgs coupling to muons, this approximation can now be dropped. However, when including the observed branching ratio to muons, we will not be sensitive to the sign of the muon Yukawa, except for the fact that such a sign flip is not consistent with the SMEFT assumptions.

Other single-current operators modify gauge and gauge-Higgs ($HVff$) couplings [4,36–42],

$$\begin{aligned}
\mathcal{O}_{\phi u}^{(1)} &= \phi^\dagger(i\overleftrightarrow{D}_\mu\phi)(\bar{u}_R\gamma^\mu u_R) &\qquad \mathcal{O}_{\phi Q}^{(1)} &= \phi^\dagger(i\overleftrightarrow{D}_\mu\phi)(\bar{Q}\gamma^\mu Q) \\
\mathcal{O}_{\phi d}^{(1)} &= \phi^\dagger(i\overleftrightarrow{D}_\mu\phi)(\bar{d}_R\gamma^\mu d_R) &\qquad \mathcal{O}_{\phi Q}^{(3)} &= \phi^\dagger(i\overleftrightarrow{D}_\mu^a\phi)\left(\bar{Q}\gamma^\mu\frac{\sigma_a}{2}Q\right) \\
\mathcal{O}_{\phi e}^{(1)} &= \phi^\dagger(i\overleftrightarrow{D}_\mu\phi)(\bar{e}_R\gamma^\mu e_R)\,.
\end{aligned} \tag{5}$$

The four-lepton operator

$$\mathcal{O}_{4L} = (\bar{L}_1 \gamma_\mu L_2)(\bar{L}_2 \gamma^\mu L_1) \tag{6}$$

induces a shift in the Fermi constant. For the operators in Eq.(5), we maintain for simplicity a flavor symmetry, and all currents are implicitly defined with diagonal flavor indices. In this limit, the operators $\mathcal{O}^{(1)}_{\phi L}, \mathcal{O}^{(3)}_{\phi L}$, analogous to $\mathcal{O}^{(1)}_{\phi Q}, \mathcal{O}^{(3)}_{\phi Q}$, are redundant with the bosonic set of Eq.(3) via equations of motion [3,29].

Dipole operators and $\mathcal{O}^{(1)}_{\phi ud,ij} = \tilde{\phi}^\dagger (i D_\mu \phi)(\bar{u}_{R,i}\gamma^\mu d_{R,j})$ are neglected for two reasons: the approximate flavor symmetry requires them to scale with the SM Yukawa couplings and their interference with the SM is always proportional to the fermion masses. Both factors suppress their effects except for the top quark. The three dipole moments of the top quark — electric, magnetic and chromomagnetic — are not suppressed, so in this work we choose to retain the chromomagnetic operator [4,43–45]

$$\mathcal{O}_{tG} = i g_s (\bar{Q}_3 \sigma^{\mu\nu} T^A u_{R,3}) \, \tilde{\phi} \, G^A_{\mu\nu} \,. \tag{7}$$

It affects the Higgs observables at the LHC significantly through the loop-induced production process [12,13,46–48].

Our SMEFT Lagrangian is then defined as

$$
\begin{aligned}
\mathcal{L}_{\text{eff}} = \mathcal{L}_{\text{SM}} &- \frac{\alpha_s}{8\pi}\frac{f_{GG}}{\Lambda^2}\mathcal{O}_{GG} + \frac{f_{WW}}{\Lambda^2}\mathcal{O}_{WW} + \frac{f_{BB}}{\Lambda^2}\mathcal{O}_{BB} + \frac{f_{BW}}{\Lambda^2}\mathcal{O}_{BW} \\
&+ \frac{f_W}{\Lambda^2}\mathcal{O}_W + \frac{f_B}{\Lambda^2}\mathcal{O}_B + \frac{f_{\phi 1}}{\Lambda^2}\mathcal{O}_{\phi 1} + \frac{f_{\phi 2}}{\Lambda^2}\mathcal{O}_{\phi 2} + \frac{f_{3W}}{\Lambda^2}\mathcal{O}_{3W} \\
&+ \frac{f_\mu m_\mu}{\nu\Lambda^2}\mathcal{O}_{e\phi,22} + \frac{f_\tau m_\tau}{\nu\Lambda^2}\mathcal{O}_{e\phi,33} + \frac{f_b m_b}{\nu\Lambda^2}\mathcal{O}_{d\phi,33} + \frac{f_t m_t}{\nu\Lambda^2}\mathcal{O}_{u\phi,33} \\
&+ \frac{f^{(1)}_{\phi Q}}{\Lambda^2}\mathcal{O}^{(1)}_{\phi Q} + \frac{f^{(1)}_{\phi d}}{\Lambda^2}\mathcal{O}^{(1)}_{\phi d} + \frac{f^{(1)}_{\phi u}}{\Lambda^2}\mathcal{O}^{(1)}_{\phi u} + \frac{f^{(1)}_{\phi e}}{\Lambda^2}\mathcal{O}^{(1)}_{\phi e} + \frac{f^{(3)}_{\phi Q}}{\Lambda^2}\mathcal{O}^{(3)}_{\phi Q} + \frac{f_{4L}}{\Lambda^2}\mathcal{O}_{4L} \\
&+ \frac{f_{tG}}{\Lambda^2}\mathcal{O}_{tG} + \text{invisible decays}\,. 
\end{aligned}\tag{8}
$$

It contains 20 independent Wilson coefficients. The branching ratio of the Higgs to invisible final states, $\text{BR}_{\text{inv}}$, is treated as a free parameter, to account for potential Higgs decays to a dark matter agent. For the global analysis it is convenient to work with the two orthogonal combinations

$$\mathcal{O}_\pm = \frac{\mathcal{O}_{WW} \pm \mathcal{O}_{BB}}{2} \qquad \Rightarrow \qquad f_\pm = f_{WW} \pm f_{BB}\,. \tag{9}$$

The rotation is defined such that only $\mathcal{O}_+$ contributes to the $H\gamma\gamma$ vertex.

If we base our calculation on the Lagrangian like that given in Eq.(8), we strictly speaking need to supplement it with a renormalization scheme or a renormalization condition. For each process, a reasonable assumption is that all Lagrangian parameters, including the Wilson coefficients, are evaluated at the same renormalization scale $\mu_R$. For the processes entering our global analysis, an appropriate central scale choice is $\mu_R \simeq m_H/2 \ldots m_H$. To improve the precision beyond leading order, one should eventually account for the renormalization group evolution [49], and evaluate the SMEFT predictions at the energy scale appropriate for each process. This scale can vary for instance across bins of a kinematic distribution. In this work, all SMEFT predictions are calculated at leading order, so we postpone an in-depth analysis of renormalization group effects to a future work, together with a systematic study of the impact of higher-order corrections to inclusive Higgs production and decay rates.

The truncated Lagrangian of Eq.(8) as our fundamental theory hypothesis needs to be put into context. The hypothesis based on a truncated Lagrangian is, strictly speaking, not well defined once we include higher multiplicities of the dimension-6 operators in the amplitude. Therefore, the SMEFT analysis should be interpreted as representing classes of models [50,51], and the validity of the SMEFT approach rests on the process-dependent assumption that in the corresponding models no new particle is produced on its mass shell [52]. While SMEFT is an excellent framework to interpret global LHC analysis, possible anomalies need to be interpreted by matching it to UV-complete models [53–56], where for instance WBS signatures of corresponding models might eventually require us to go beyond dimension-6 operators [57].

If global SMEFT analyses should be interpreted as representing classes of UV-complete models for a limited set of observables, we need to consider the interplay between the SMEFT hypothesis and more fundamental models. Given the precision of the SMEFT analysis and its field-theoretical advantages over the naive coupling analysis we can and should perform this matching beyond leading order [58–63], accounting for matching scale uncertainties [7,64], rather than ignoring them at leading order. While this scale uncertainty clearly does not cover all uncertainties induced by matching SMEFT limits to UV-complete models, it also illustrates that such uncertainties exist and have to be taken into account.

## 3   Bayesian SFitter setup

Global SMEFT analyses are a key ingredient to a more general analysis strategy at the LHC, which is to test theory predictions based on perturbative quantum field theory using the full kinematic range of the complete set of LHC measurements. It is worth stressing that SMEFT analyses are currently the only way to systematically probe kinematic LHC measurements beyond resonance searches. They come with two assumptions which greatly simplify the actual analyses

1. experimentally, we know that our SMEFT analysis is not confronted with established anomalies; those should be discussed using properly defined BSM models;
2. theoretically, SMEFT can only describe small deviations from the Standard model, otherwise the dimensional expansion in Eq.(1) is not valid.

While global SMEFT analyses with a truncated Lagrangian can translate kinematic measurements into fundamental parameters, these two aspects imply that their outcome will be limit-setting. For our analysis this means that we already know that the global maximum of the SMEFT likelihood lies around the SM-limit $f_x/\Lambda^2 \to 0$. The exact position of the most likely parameter point is of limited interest, the main task of the global analysis is to determine the uncertainty on the values of the Wilson coefficients or, more in general, the finite preferred region in the multi-dimensional SMEFT parameter space.

In this spirit, the goal of the SFitter framework is to enable an independent interpretation of experimental inputs, without relying on pre-processed information and including a comprehensive treatment of statistical, systematic, and theory uncertainties [16–18]. The SFitter methodology relies on the construction of a likelihood function in which these uncertainties can be described by nuisance parameters. In all previous SFitter analyses, nuisance parameters are profiled over. The resulting profile likelihood is then profiled over the parameters of interest, to extract one- and two-dimensional limits on the Wilson coefficients. An alternative, Bayesian treatment is based on marginalising over nuisance parameters and parameters of interest. It has been adopted in several SMEFT analyses [15,65–68] and simplifies greatly the treatment of correlated uncertainties. The goal of this work is to perform an apples-to-apples comparison between a profiled and a marginalised likelihood, employing exactly the same data and uncertainties inputs in both cases.

**Marginal likelihood**

Since marginalization is new in SFitter, we provide a brief description of the main features. The corresponding profile likelihood treatment is discussed in detail in Refs. [3, 11, 16, 17, 19]. The first step of a global analysis is the construction of the fully exclusive likelihood $\mathcal{L}_{\text{excl}}$, which is a function of the parameters of interest $f_x$ and of a set of nuisance parameters $\theta_i$. This $\mathcal{L}_{\text{excl}}$ is defined with the following uncertainty treatment: (i) statistical uncertainties are included via a Poisson distribution, in some cases approximated using a Gaussian whenever this stabilizes the numerical evaluation; (ii) systematic uncertainties are assumed to be Gaussian, organized in 31 categories, such that uncertainties within the same category are fully correlated through a covariance matrix or through nuisance parameters. Systematics which do not fit into any of the 31 categories are assumed to be uncorrelated; (iii) theory uncertainties are modelled as flat distributions. Whenever theory uncertainties need to be correlated we use an explicit nuisance parameter.

For a Bayesian analysis we first marginalize over or integrate out the nuisance parameters. This yields the marginal likelihood $\mathcal{L}_{\text{marg}}$, for one counting measurement and one parameter illustrated by

$$\mathcal{L}_{\text{marg}}(f_x) = \int d\theta \, \mathcal{L}_{\text{excl}}(f_x, \theta) = \int d\theta \, \text{Pois}(d|m(f_x, \theta)) \, p(\theta) \,. \tag{10}$$

Here $d$ stands for the measured number of events, $m$ is the model (theory) prediction, $\theta$ is a nuisance parameter and $p(\theta)$ the distribution over the nuisance parameter which, in the Bayesian context, defines the prior. In SFitter, nuisance priors are either Gaussian or flat. Computing $\mathcal{L}_{\text{marg}}$ in SFitter starts with the marginalisation procedure over the nuisance parameters, so we omit the dependence on $f_x$ for now.

SFitter provides several options to define the statistical model of a measurement, including a simplified Gaussian likelihood where uncertainties add in quadrature. A more sophisticated and reliable framework starts with a typical LHC measurement as an independent counting experiment, which is modelled by a Poisson distribution. Systematic uncertainties or theory uncertainties then define the completely exclusive likelihood for one measurement

$$\mathcal{L}_{\text{excl}}(\theta) = \text{Pois}(d|m(\theta_1, \theta_2, ..., b)) \, p(b) \prod_i p(\theta_i) \,. \tag{11}$$

Here $d$ is the measured number of events, $b$ the background estimate, and $m$ the model prediction, that is a function of the nuisance parameters $\theta_i$. The distributions $p(b)$ and $p(\theta_i)$ incorporate our knowledge about these quantities. In general, they can be extracted from auxiliary measurements, simulations, or other possible sources. However, because tracking hundreds of different reference measurements is beyond the scope of SFitter, we simply assume $p(\theta_i)$ to be Gaussian for systematic uncertainties and flat or uniform for theory uncertainties,

$$p(\theta_i) = \begin{cases} \mathcal{N}_{0,\sigma_i}(\theta_{\text{syst},i}) & \text{systematics} \\ \mathcal{F}_{0,\sigma_i}(\theta_{\text{theo},i}) & \text{theory} \,. \end{cases} \tag{12}$$

In this step we assume that all prior distributions for $\theta_{\text{syst}}$ and $\theta_{\text{theo}}$ are centered around zero, with given half-widths $\sigma$.

For $p(b)$, SFitter provides several choices: for measurements where $b$ is extracted from a single control region (CR) measurement we use

$$p(b) = \text{Pois}(b_{\text{CR}}|bk) \,, \tag{13}$$

where $k$ is an interpolation factor between CR and signal region, $b_{CR}$ is the measured number of events in the control region, and $b$ is the expected number of background events in the signal region. For measurements with several control regions or with simulated backgrounds we assume the combined $p(b)$ to be a Gaussian. Systematic uncertainties on the background measurement can also be included, and are assumed to be fully correlated with the uncertainties on the signal region within the same category.

Typically, the dependence of the theory prediction $m$ on the nuisance parameters in Eq.(11) is not spelled out or extremely complex to determine. To simplify this task, we assume a leading linear dependence on assumed-to-be small uncertainties

$$m \approx s + b + \theta_{\text{theo},1} + \theta_{\text{theo},2} + \cdots + \theta_{\text{syst},1} + \theta_{\text{syst},2} + \cdots \equiv s + b + \theta_{\text{tot}} \,. \tag{14}$$

where $s$ is the expected number of signal events. The exclusive likelihood of Eq.(11) can then be written as

$$\mathcal{L}_{\text{excl}}(\theta) \approx \text{Pois}(d|s + b + \Sigma\theta_{\text{theo},j} + \Sigma\theta_{\text{syst},i}) \, p(b) \prod_j \mathcal{F}_{0,\sigma_j}(\theta_{\text{theo},j}) \prod_i \mathcal{N}_{0,\sigma_i}(\theta_{\text{syst},i}) \,, \tag{15}$$

The marginal likelihood for a single measurement is then constructed by integrating over all nuisance parameters,

$$\begin{aligned}
\mathcal{L}_{\text{marg}} &= \int \prod_j d\theta_{\text{theo},j} \int \prod_i d\theta_{\text{syst},i} \int db \, \mathcal{L}_{\text{excl}}(\theta) \\
&= \int \prod_j d\theta_{\text{theo},j} \mathcal{F}_{0,\sigma_j}(\theta_{\text{theo},j}) \int \prod_i d\theta_{\text{syst},i} \mathcal{N}_{0,\sigma_i}(\theta_{\text{syst},i}) \\
&\qquad\qquad \times \int db \, \text{Pois}(d|s + b + \Sigma\theta_{\text{theo},j} + \Sigma\theta_{\text{syst},i}) \, p(b) \,. \tag{16}
\end{aligned}$$

The integration over $b$ can be performed analytically if $p(b)$ is a Poisson distribution. In this case, the convolution $\mathcal{P}(d|s + \theta_{\text{tot}})$ of $p(b)$ and $\text{Pois}(d|m)$ gives a so-called Poisson-Gamma model, as Eq.(13) is a special case of the Gamma distribution,

$$\mathcal{L}_{\text{marg}} = \int \prod_j d\theta_{\text{theo},j} \mathcal{F}_{0,\sigma_j}(\theta_{\text{theo},j}) \int \prod_i d\theta_{\text{syst},i} \mathcal{N}_{0,\sigma_i}(\theta_{\text{syst},i}) \times \mathcal{P}(d|s + \theta_{\text{tot}}) \,. \tag{17}$$

We use $\theta_{\text{tot}}$ as defined in Eq.(14). To solve the remaining integrals over the nuisance parameters we replace one of the integrals, for instance $\theta_{\text{syst},1}$ with $(\theta_{\text{tot}} - \Sigma_{i \neq 1}\theta_{\text{syst},i})$,

$$\begin{aligned}
\mathcal{L}_{\text{marg}} = \int d\theta_{\text{tot}} \, \mathcal{P}(d|s + \theta_{\text{tot}}) \\
\times \underbrace{\int \prod_j d\theta_{\text{theo},j} \mathcal{F}_{0,\sigma_j}(\theta_{\text{theo},j}) \int \prod_{i \neq 1} d\theta_{\text{syst},i} \mathcal{N}_{0,\sigma_i}(\theta_{\text{syst},i}) \mathcal{N}_{0,\sigma_1}(\theta_{\text{syst},1})}_{\text{solved analytically}} \,. \tag{18}
\end{aligned}$$

Assuming only Gaussian plus at most three flat priors, all $\theta$-convolutions except for one can be performed analytically. The corresponding closed formulas are implemented in SFitter, speeding up the marginalisation. The remaining 1-dimensional integral in Eq.(18) is solved numerically with Simpson's method.

Marginalizing over nuisance parameters and profiling over them will not give the same marginalized likelihood. Only for statistical uncertainties described by Poisson statistics and

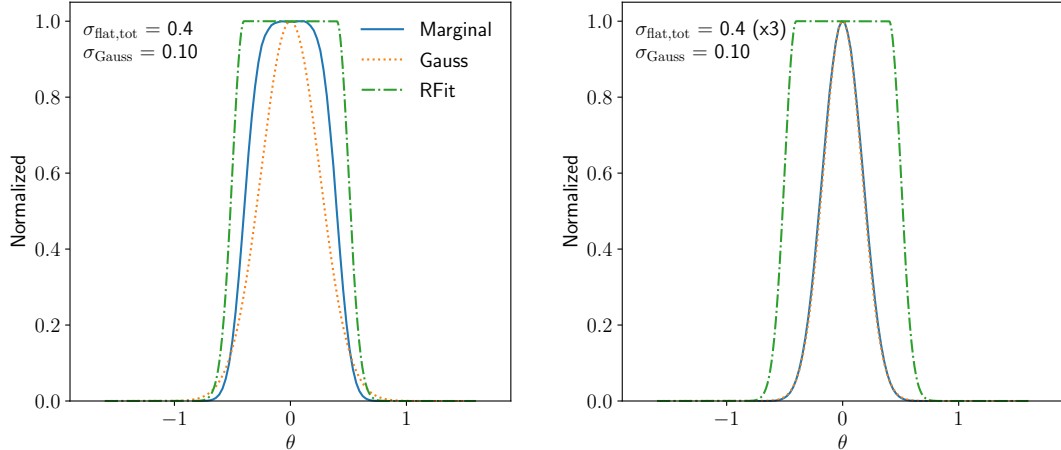

Figure 1: Marginalized and profiled likelihoods from the convolution of a Gaussian distribution with one (left) and three (right) flat ones. The orange curve shows, for comparison, the Gaussian obtained adding half-widths in quadrature.

Gaussian systematics, the two lead to the same marginalized result in the limit of large enough statistics. Differences appear when we use flat theory uncertainties. For a Bayesian marginalization the central limit theorem ensures that the final posterior will be approximately Gaussian. Using a profile likelihood, two uncorrelated flat uncertainties add linearly, while a combination of flat and Gaussian uncertainties give the well-known RFit prescription [69]. Figure 1 shows, as an illustration, the distributions obtained combining one Gaussian with one (left) or three (right) flat nuisance parameters. We see that the profile likelihood or RFit result maintains a flat core and is independent of the number of theory nuisances, while the marginalised result varies and is very close to a Gaussian in the right panel.

**Combining channels**

Unlike probabilities, likelihoods of a set of measurements can simply be multiplied. This means we can generalize Eqs.(11) and (15) to a set of $N$ measurements by replacing

$$\text{Pois}(d|m)p(b) \quad \longrightarrow \quad \prod_k \text{Pois}(d_k|m_k)p(b_k)$$

$$\mathcal{N}_{0,\sigma_i}(\theta_{\text{syst},i}) \quad \longrightarrow \quad \mathcal{N}_{\vec{0},\Sigma_i}(\vec{\theta}_{\text{syst},i})$$

$$\mathcal{F}_{0,\sigma_j}(\theta_{\text{theo},j}) \quad \longrightarrow \quad \prod_k \mathcal{F}_{0,\sigma_{kj}}(\theta_{\text{theo},kj}) \,, \tag{19}$$

with

$$m_k \approx s_k + b_k + \sum_i \theta_{\text{syst},ki} + \sum_j \theta_{\text{theo},kj} \equiv s_k + b_k + \theta_{\text{tot},k} \,. \tag{20}$$

Here we assume that the theory uncertainties are uncorrelated, while the systematics can be correlated, so we need to introduce an $N$-dimensional Gaussian with the covariance matrices $\Sigma_i$ encoding the correlations between uncertainties of category $i$ entering different measurements $k$. We use either uncorrelated or fully correlated systematics.

When we compute the marginal likelihood in analogy to Eq.(16) the only non-trivial aspect are the correlated systematic uncertainties including the covariance matrix. However, the convolution of $N$-dimensional Gaussians still leads to one $N$-dimensional Gaussian, where the

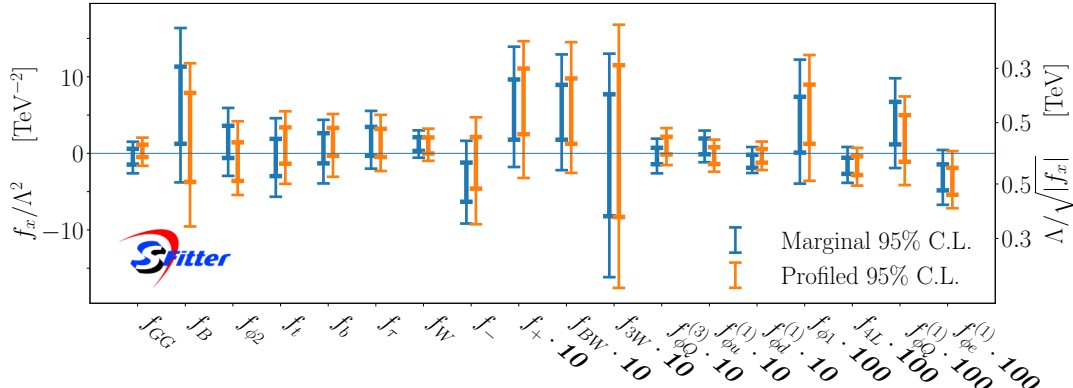

Figure 2: 68% and 95% confidence intervals from profile likelihoods and Bayesian marginalization. The dataset is the same as in Ref. [3].

combined covariance matrix is the sum of the individual covariance matrices. This means, in the last step of Eq.(18) we are now left with an $N$-dimensional integral over $\theta_{\text{tot},k}$, correlated through the covariance matrix appearing in the distribution of the systematic nuisance parameters.

In SFitter, this integral is solved by approximating it with the Laplace method. This is computationally efficient and works well for cases where most of the probability is concentrated around one mode. This is the case when the nuisance parameters are Gaussians or flat. We can then write

$$\int dx^n f(x) = \int dx^n e^{\log f(x)}, \tag{21}$$

and assume that $f(x)$ has a maximum at $x = x_0$. Then one can expand $\log f(x)$ up to second order around $x_0$ as

$$\log f(x) \approx \log f(x_0) + \underbrace{\frac{\partial}{\partial x} \log f(x_0)(x - x_0)}_{=0} + \underbrace{\frac{\partial^2}{\partial x_i x_j} \log f(x_0)(x - x_0)_i (x - x_0)_j}_{=F_{ij}(x_0)} + \dots \tag{22}$$

such that the integral is approximated by

$$\int dx^n f(x) \approx f(x_0) \sqrt{\frac{(2\pi)^n}{\det F(x_0)}}. \tag{23}$$

Note that $f(x)$ is given by the exclusive likelihood, with the maximum at $f(x_0)$ kept through profiling but not through marginalization. The matrix $F(x_0)$ is the Hessian of the log-likelihood at the maximum, *i.e.* the Fisher information matrix in the space of the nuisance parameters. In SFitter, $x_0$ is extracted with an analytic expression, approximating the Poisson distribution in Eq.(11) with a Gaussian. The resulting error is compensated by keeping a finite first derivative in Eq.(22), which in turn requires us to modify Eq.(23) by introducing an additional term depending on the first derivative of the log-likelihood. Both the first and second derivatives can be computed numerically. All these approximations in evaluating the exclusive and marginal likelihoods have been checked by evaluating the exclusive likelihood using Markov chains.

**Validation**

We can validate the implementation of the Bayesian marginalization over nuisance parameters and Wilson coefficients starting from the fully exclusive likelihood using the operator basis

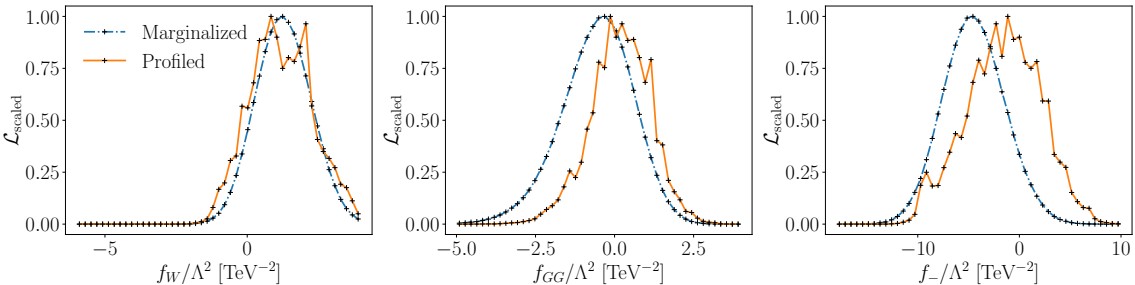

Figure 3: Profile likelihoods vs marginalized likelihood for a set of Wilson coefficients. The two curves are scaled such that the maximum values are at $\mathcal{L}_{\text{scaled}} = 1$.

and dataset of Ref. [3]. The SMEFT Lagrangian is given in Eq.(8), but without the muon Yukawa, the top-gluon coupling $\mathcal{O}_{tG}$, and the invisible branching ratio of the Higgs. For the direct comparison we construct the marginal likelihood by profiling or marginalizing over all nuisance parameters and Wilson coefficients. We then extract the posterior probability and 68% and 95% confidence intervals. Unless otherwise specified, we assume flat, wide priors for all Wilson coefficients. This choice minimizes the impact of the prior on the final result, and we have verified that our priors on the Wilson coefficients indeed fulfill this condition. In Fig. 2, we show the 68% and 95%CL limits from the corresponding 18-dimensional operator analysis. We see that the results of the two methods are in excellent agreement.

Going beyond confidence intervals, we can look at the distributions of the 1-dimensional profile likelihoods or marginalized probabilities. We show three examples in Fig. 3. Because the analysis relies on actual LHC data, the central values are not at zero Wilson coefficients. The well-measured Wilson coefficient $f_W$ shows no difference between the profile and the marginalized results. For $f_{GG}$, we see a slight deviation in the central values, within one standard deviation and therefore not statistically significant. This effect points to the theory and pdf uncertainties, which we assume to be flat, and which therefore allow the central value to move freely for the profile likelihood approach, while the marginalization leads to a well-defined maximum when combining two individually flat likelihood distributions. In Fig. 2 we see that this difference only has a slight effect on the lower boundary when we extract 95%CL limits on $f_{GG}$. Finally, we see a similar effect for $f_-$, even though this measurement depends on several different LHC channels. According to Fig. 2 this is one of the largest and still not significant differences between the two methods.

The source of the differences in Fig. 3 can be traced back to whether the uncertainty-related nuisance parameters are marginalised or profiled. Fig. 4 shows that, once the uncertainty treatment is fixed, the results are independent of whether the Wilson coefficients are marginalized

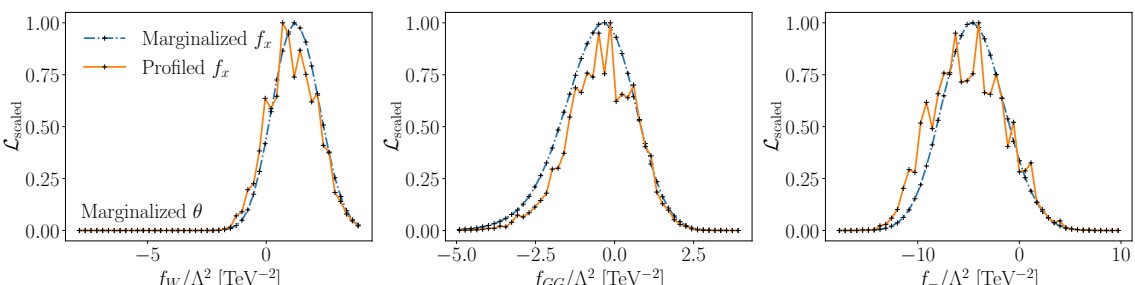

Figure 4: Likelihoods profiled vs marginalized over the Wilson coefficients $f_x$, but always marginalized over all nuisance parameters $\theta$. We show the same Wilson coefficients as in Fig. 3.

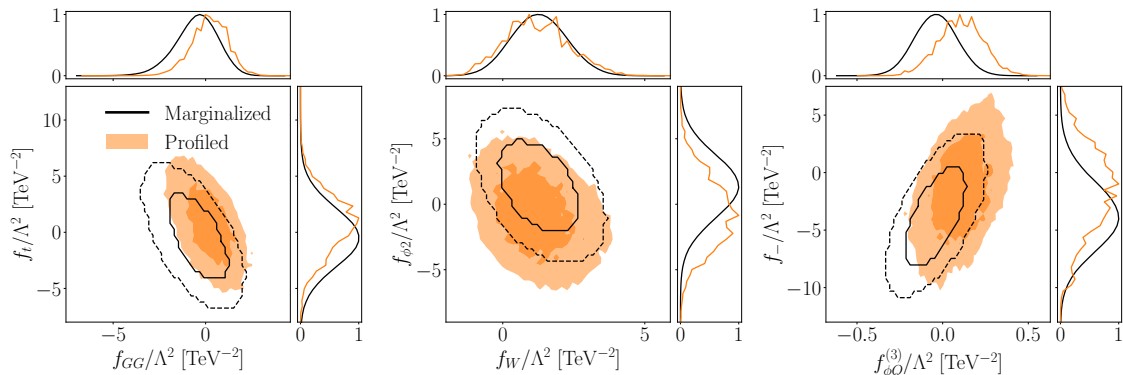

Figure 5: Comparison of 2-dimensional correlations of profiled and marginalized likelihoods.

or profiled over.

Next, we check 2-dimensional profiled and marginalised likelihoods. Figure 5 shows three examples involving the same Wilson coefficients as in Fig. 3. First, we see that there exists an anti-correlation between $f_{GG}$ and $f_t$, the modified top Yukawa also affecting the loop-induced production process $gg \to H$. This suggests that a slightly high rate measurement can be accommodated by adjusting either of the two Wilson coefficients. Because the uncertainty on this measurement includes sizeable theory and pdf contributions, the same difference between the two methods can be seen for each of the two Wilson coefficients individually and for their correlation. Another instructive example is the correlation between $f_W$, determined from kinematic distributions, and $f_{\phi 2}$ leading to a shift in the Higgs wave function. Here the difference only appears in $f_{\phi 2}$, the parameter extracted from total rates and especially sensitive to theory uncertainties. Finally, we show the correlation between $f_-$ and $f_{\phi Q}^{(3)}$ and observe the usual correlation from the sizeable range of kinematic di-boson measurements [70].

Finally, we can check for alternative maxima in the likelihood and find that $f_+$ is the only Wilson coefficient exhibiting a non-trivial second mode. This can be understood from the $f_+$ vs $f_-$ plane. By a numerical accident, the SMEFT corrections to all Higgs production and decay processes vanish in the SM-maximum and also close to the point $f_-/\Lambda^2 = -3$ and $f_+/\Lambda^2 = 2.7$. The only measurement which breaks this degeneracy is $H \to Z\gamma$, with limited statistical power. In the $f_+$ axis, the position of the maximum is fully determined by $H \to \gamma\gamma$, which is measured precisely enough to resolve the two modes, while in the $f_-$ axis the constraints cannot distinguish the second maximum from the SM point.

Given the consistency condition of the SMEFT approach, we should not compare the two modes at face value, even though the Bayesian setup would allow for this. On the other hand,

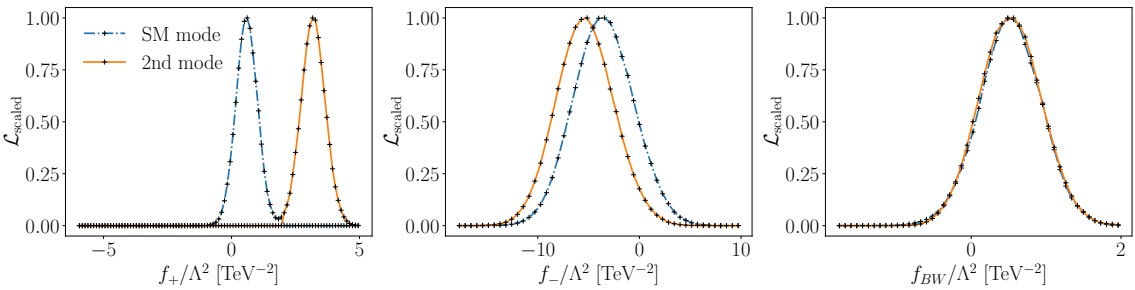

Figure 6: Marginalized likelihoods for the SM-like and the second mode in $f_+$, again for the 18-dimensional analysis.

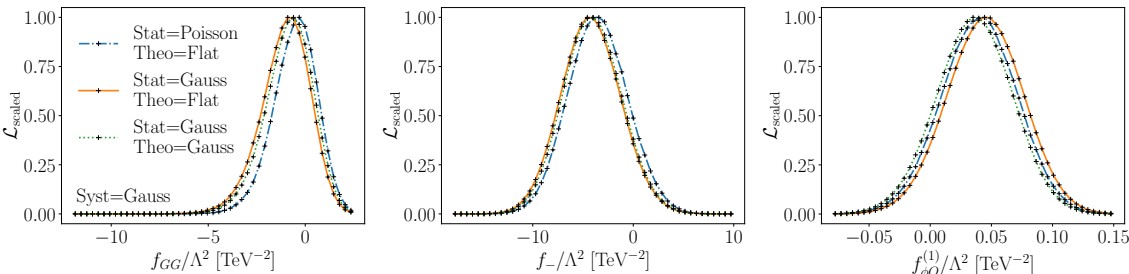

Figure 7: Marginalized likelihoods for different uncertainty modeling. The SFitter default is a Poisson likelihood with flat theory uncertainties and Gaussian systematics (blue dot-dashed).

we need to confirm that this choice of modes does not affect other parameters in a significant manner once it is embedded in the 18-dimensional space. In Fig. 6 we show what happens if we restrict our parameter analysis to either the SM-mode or the second mode. To this end we run Markov chains mapping out both modes and then separate the samples through the condition $f_+/\Lambda^2 \lesssim 2$. We see that choosing the second mode in $f_+$ has a small effect on $f_-$, pushing the best-fit closer to $f_- = -3$, but none of the other Wilson coefficients is affected. We also confirmed that both modes are of equal height by choosing a Breit-Wigner proposal function, which ensures that the Markov chains can move large distances, helping each individual chain to jump between both modes.

**Uncertainties and correlations**

After confirming that the slight differences between the profile and marginalization approaches are related to the treatment of uncertainties, we can check the impact of the SFitter-specific uncertainty treatment. By default, and as explained earlier, we construct the exclusive likelihood with flat theory uncertainties and Gaussian systematics. By switching all uncertainties to Gaussian distributions we construct the completely Gaussian likelihood shown in Fig. 7. If we marginalize over the different uncertainties, the central limit theorem guarantees that for enough different uncertainties the results will be identical. The exact level of agreement between different uncertainty models depends on the dataset and the size of the individual uncertainties and cannot be generalized. For instance, sizeable differences will appear when an outlier measurement generates a tension in the global analysis. Such a tension can be accommodated more easily using a single flat uncertainty with its reduced cost in the likelihood value.

Because the main difference between profiling and marginalizing over uncertainties appears for the flat theory uncertainties, the results from Fig. 7 motivate the question how rel-

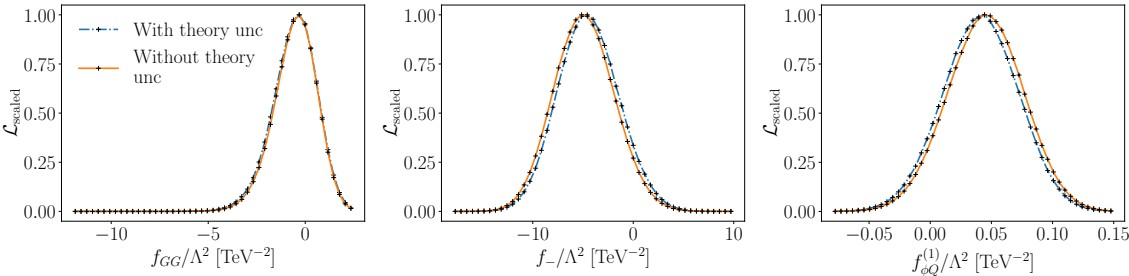

Figure 8: Marginalized likelihoods with and without theory uncertainties.

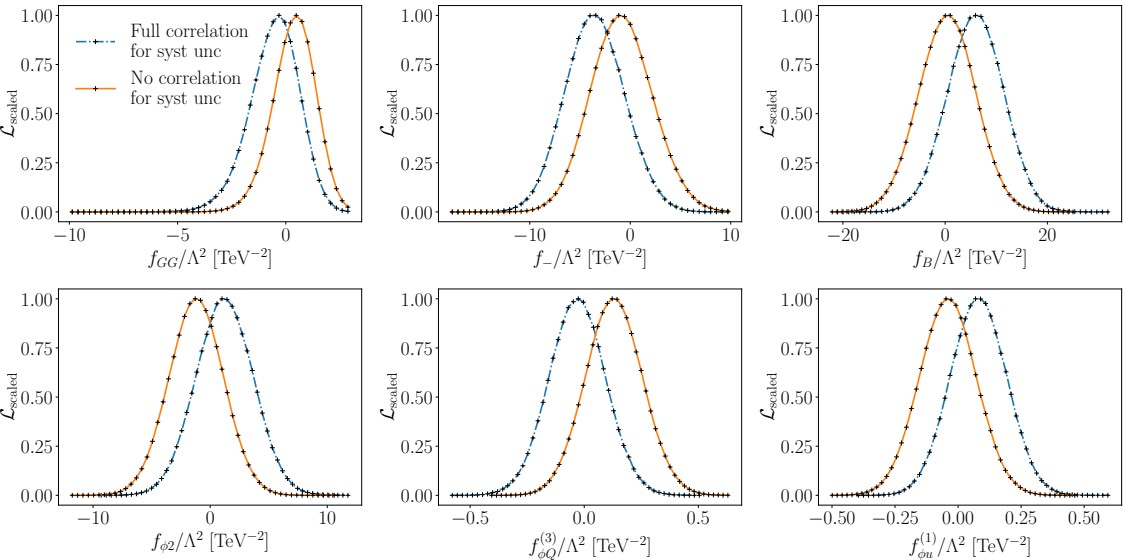

Figure 9: Marginalized likelihoods with and without correlations between systematic uncertainties of the same category.

evant the theory uncertainties really are for the Run 2 dataset analyzed in Ref. [3]. We show three 1-dimensional likelihoods in Fig. 8 and indeed find that after marginalizing over all nuisance parameters and over all other Wilson coefficients the theory uncertainties do not play any visible role. Obviously, this statement is dependent on a given dataset, on the operators we are looking at, and on the assumed uncertainties, and it clearly does not generalize to all global Run 2 analyses.

The last effect we need to study is the impact of correlations between the different uncertainties. In Fig. 9 we show what happens with the 1-dimensional marginalized likelihoods when we switch off all correlations between systematic uncertainties of the same kind. We see that the correlations have a much larger impact than anything else we have studied in this section. While the size of the uncertainties do not change much, the central values essentially vary freely within one standard deviation. An analogous effect was observed in Ref. [15]. We cannot emphasize enough that all statements about the validity of different approximations do not generalize to new, incoming measurements, as we will see in the following section. However, something that will not change is the key relevance of correlations as indicated by Fig. 9.

# 4   Updated dataset

After the detailed comparison of a profile likelihood and Bayesian SFitter approach we can, in principle, apply the numerically simpler Bayesian approach to update the SMEFT analysis of the Higgs-electroweak sector with a series of new Run 2 results. As a first step, we introduce the set of new kinematic measurements entering the updated SFitter analysis. We focus on an improved treatment of correlated uncertainties.

## 4.1   WW resonance search

Once we notice that especially boosted kinematics with large momentum transfer through Higgs interactions play a key role in SMEFT analyses [70, 72], it is clear that the reinter-

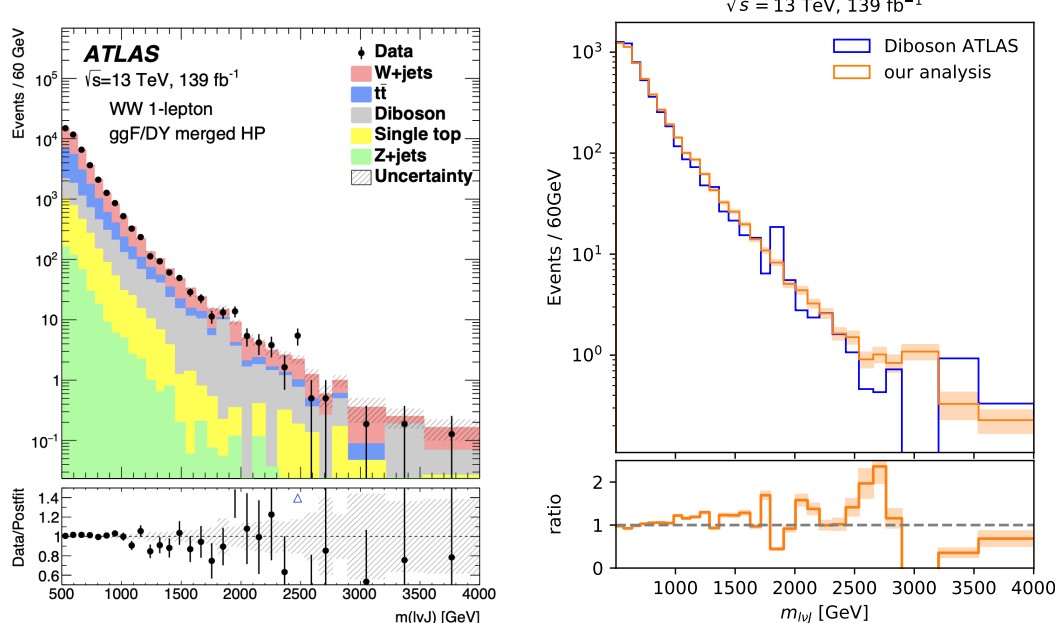

Figure 10: Left: measured $m_{VV}$ distribution [71]. Right: comparison between AT-LAS results and our SM background estimate. The orange band shows the statistical uncertainty from the Monte Carlo generation.

pretation of $VH$ and $VV$ resonance searches should be extremely useful for a global SMEFT analysis [3, 19]. To the best of our knowledge, SFitter is currently the only global analysis framework which includes these kinds of signatures.

First, we add the ATLAS search for resonances in the semi-leptonic $VV$ final state [71], as briefly discussed in Ref. [64]. We only use the $WW$ 1-lepton category in the merged Drell-Yan and gluon-fusion high-purity signal region,

$$pp \to W^+ W^- \to \ell^+ \nu_\ell \, jj + \ell^- \bar{\nu}_\ell \, jj \,. \qquad (24)$$

Our signal consists of $W^+W^-$ production modified by SMEFT operators. We neglect SMEFT effects in the leading $W$+jets and $t\bar{t}$ backgrounds. We include all other $W_{\ell\nu}V_{jj}$ and $Z_{\ell\ell}V_{jj}$ channels as SM-backgrounds and verified that SMEFT corrections to the other di-boson channels are sufficiently suppressed by the analysis setup.

The signal is simulated using Madgraph [73], Pythia [74], FastJet [75], and Delphes [76] with the standard ATLAS card at leading order and in the SM and requiring the lepton pair to come from an intermediate on-shell $W^\pm$. The hadronic $W$-decay is simulated using Pythia. Fat jets are identified using the default categorization in Delphes and ignoring the cut on the $D_2$ variable. The complete SM-rate is compared to the left panel of Fig. 10, taken from Ref. [71]. We reproduce the event selection based on the analysis cuts listed in Tab. 2 of Ref. [71]. No re-calibration of energy scales or fat-jet invariant mass windows is required, but we adjust the histogram entries by a factor 1.606 to match the ATLAS normalization of the di-boson background and accommodate efficiencies and higher-order corrections [77]. In the right panel of Fig. 10 we show the final $m_{WW}$ distribution obtained with this procedure. Finally, we extract the statistical and systematic uncertainty from the ATLAS analysis, as shown in the lower panel in Fig. 10. Whenever backgrounds are estimated from control regions, the Gaussian systematic uncertainties are smaller than the Poisson-shaped statistical uncertainties in the signal region.

To include the $VV$ channel in our SMEFT analysis we re-bin the original distribution such that we have a minimum of five observed events per bin. The kinematic distribution we use

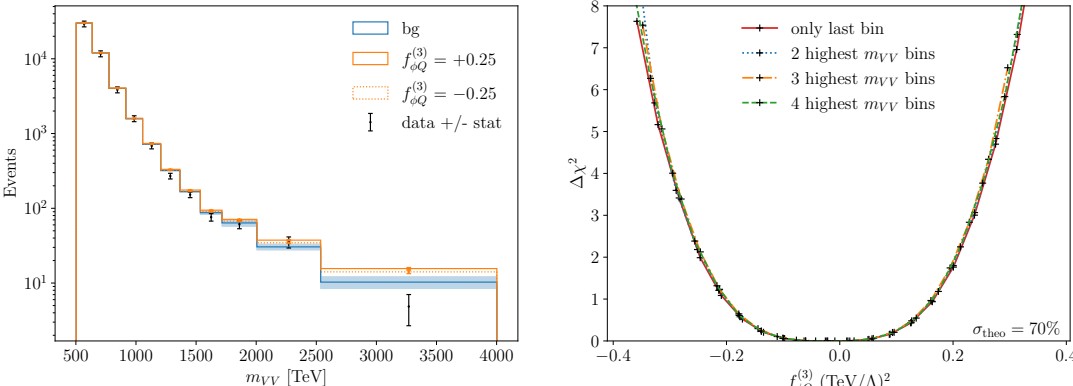

Figure 11: Left: re-binned $m_{WW}$ distribution for the semi-leptonic $WW$ analysis implemented in SFitter. We show the complete continuum background, including statistical and systematic uncertainties, and the effect of a finite Wilson coefficient $f_{\phi Q}^{(3)}$. Right: toy analysis for the same Wilson coefficient using different numbers of bins.

in SFitter is shown in the left panel of Fig. 11. Here all statistical uncertainties are treated as uncorrelated and added in quadrature, the same for the systematic background uncertainties linked to Monte Carlo statistics, while other systematic uncertainties are conservatively treated as fully correlated and consequently added linearly. Finally, we add a 80% theory uncertainty on the signal predictions in all bins and assuming no correlation among them. Of this 70% account for the uncertainties in our SMEFT Monte Carlo predictions and 10% for $V$+jets and single-top modeling.

In the right panel of Fig. 11 we show the limit in terms of the Gauss-equivalent

$$\Delta \chi^2 = \chi^2 - \chi^2_{\min} = -2 \log \mathcal{L} + 2 \log \mathcal{L}_{\max} \, , \tag{25}$$

extracted from different bins of the measured $m_{WW}$ distribution. We see that the likelihood maximum slightly deviates from the SM point $f_{\phi Q}^{(3)} = 0$, and the last bin completely dominates the likelihood distribution. This is expected for momentum-enhanced operators which modify the tails of momentum distributions, as systematically analyzed in Ref. [70]. We will discuss the effect of the under-fluctuation in the last bin in more detail in Sec. 5.1.

## 4.2 WH resonance search

Complementing the dataset of Ref. [3] we include two new resonance searches, one described in Ref. [64] and another ATLAS analysis looking for

$$pp \to WH \to \ell \, \bar{\nu}_\ell \, b \bar{b} \tag{26}$$

at high invariant masses [78]. We focus on $WH$ production with one $b$-tag, because it includes the best kinematic measurement at high $m_{VH}$. This analysis applies cuts on the $WH$ topology and requires exactly one single-$b$-tagged fat jet. In the merged category the $b$-tags are part of a fat jet.

We generate di-boson events for the combined di-boson channels with lepton-hadron decays

$$pp \to W_{\ell \nu} W_{jj}, \, W_{\ell \nu} Z_{jj}, \, Z_{\ell \ell} W_{jj}, \, Z_{\ell \ell} Z_{jj} \, , \tag{27}$$

again using the Madgraph-Pythia-FastJet-Delphes chain with the standard ATLAS card at leading order. They can be compared to the grey di-boson background in the left panel of Fig. 12,

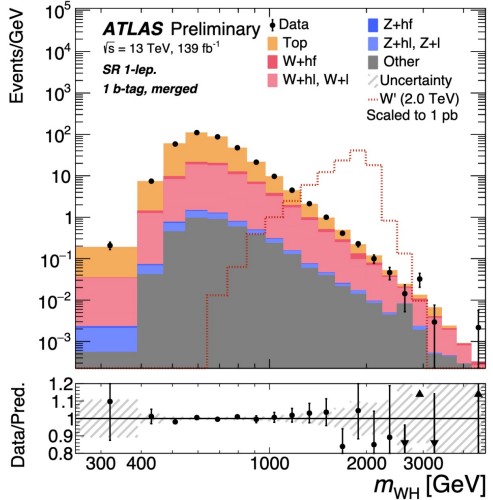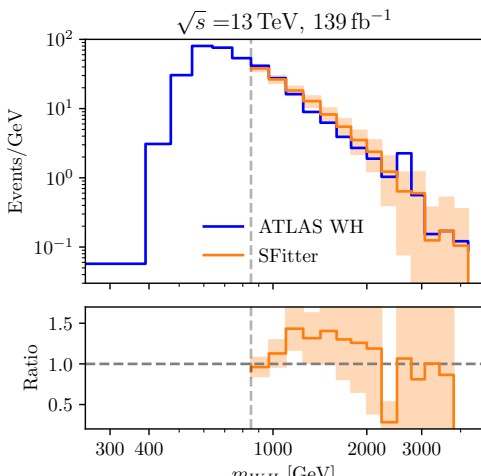

Figure 12: Left: measured $m_{WH}$ distribution [78]. Right: comparison between the the ATLAS results and our SM background estimate. The orange band shows the statistical uncertainty from the Monte Carlo generation.

including the $b$-tagging and corresponding mis-tagging. After adjusting the $m_{WH}$-independent efficiency factor we find the agreement illustrated in the right panel of Fig. 12. We apply the same efficiency factor for the $WH$ signal and then use the reweighting module in Madgraph to estimate the SMEFT rates. The $W$-decay to electrons or muons is included through Madgraph, while the Higgs decay to $b\bar{b}$ pairs is simulated by Pythia. We neglect SMEFT corrections to the $t\bar{t}$ and $W/Z$+jets backgrounds, assuming that the targeted phase space region favors the Higgs signal. Having to make this assumption is unfortunate, but we emphasize that the number of experimental measurements should prevent us from falling for SMEFT corrections canceling between the different signals and backgrounds.

To define a meaningful measurement for our global analysis we have to merge bins of the original distribution such that at least three observed events appear per bin. In Fig. 13, we show the actually implemented distribution for the complete SM background and including a finite Wilson coefficient $f_{\phi Q}^{(3)}$. For each bin we include a statistical uncertainty following a

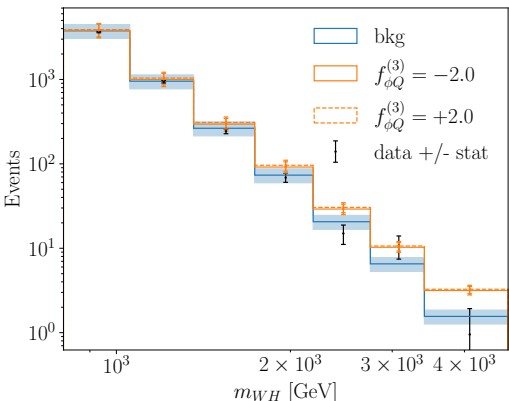

Figure 13: Re-binned $m_{WH}$ distribution implemented in SFitter, including statistical and systematic uncertainties. We show the complete continuum background and the effect of a finite Wilson coefficient $f_{\phi Q}^{(3)}$.

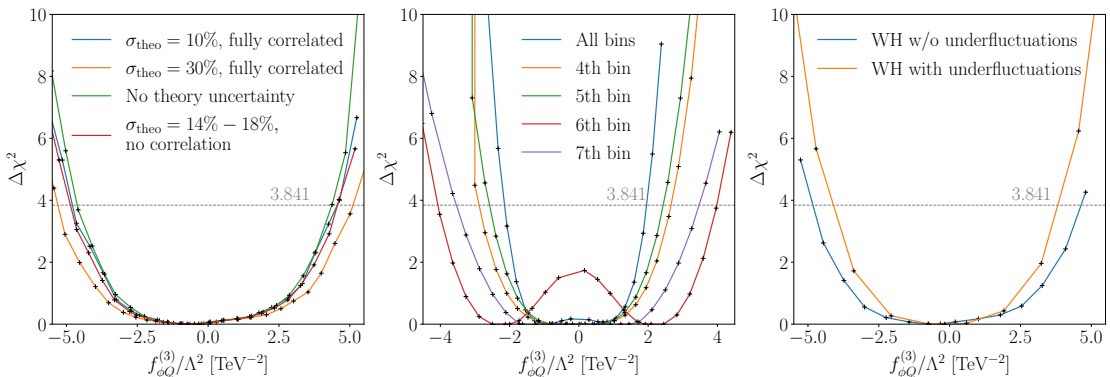

Figure 14: Log-likelihood for a 3-parameter analysis of the $WH$ search as a function of $f_{\phi Q}^{(3)}$. We vary the theory uncertainties and their correlation (left), the number of bins included with uncorrelated theory uncertainties for a 1-dimensional analysis (center), and the treatment of under-fluctuations (right).

Poisson distribution and a Gaussian systematic uncertainty, as reported by ATLAS. In addition, we include a 13% theory uncertainty also reported by ATLAS and a theory uncertainty between 1% and 4% per bin from our SMEFT predictions, but neglecting correlation between various bins.

We can check some of our assumptions on the way we model theory uncertainties from a three-parameter analysis with $f_{\phi Q}^{(3)}$, $f_W$ and $f_{WW}$. Neglecting the correlations in the theory uncertainties is justified by the left panel of Fig. 14. It shows the Gauss-equivalent $\Delta\chi^2$ for varying the theory uncertainties with different correlations; the orange and green lines represent a 10% and 30% theory uncertainty, fully correlated. The green line shows results without theory uncertainty, and the red line assumes our SMEFT theory uncertainty without correlations. These results are very close to each other, so we can ignore correlations in the theory uncertainties from the EFT prediction.

The central panel compares constraints from the 3-parameter analysis from the entire $m_{WH}$ distribution and only including one bin at a time. The limit improves sharply when the 4th and 5th bins are included. This can be understood from Fig. 13, where both of these bins show significant under-fluctuations. In the right panel of Fig. 14 we show that by removing under-fluctuations from the global analysis by setting all measured values to the number of events expected from the SM we lose constraining power. Again, demonstrating that our analysis strongly benefits from under-fluctuations.

### 4.3  ZH resonance search

The second boosted $VH$ analysis we re-interpret in terms of SMEFT is a CMS resonance search in the process [79]

$$pp \to ZH \to e^+ e^-\, b\bar{b}\,. \tag{28}$$

We include the non-VBF category with $\leq 1$ $b$-tags and with two $b$-tags. We find that the two-$b$ category is more constraining than the $\leq 1b$ category. This can happens because the relative size of the SMEFT correction prefers this category. To determine the number of $b$-tags in an event, we look at the corresponding fat jet and the number of $b$-quarks inside the jet.

We validate our analysis simulating events for $Z'$ peak in the heavy vector triplet model (HVT), that is used by CMS to illustrate a possible signal,

$$pp \to Z' \to Z_{\ell\ell} H_{bb}\,. \tag{29}$$

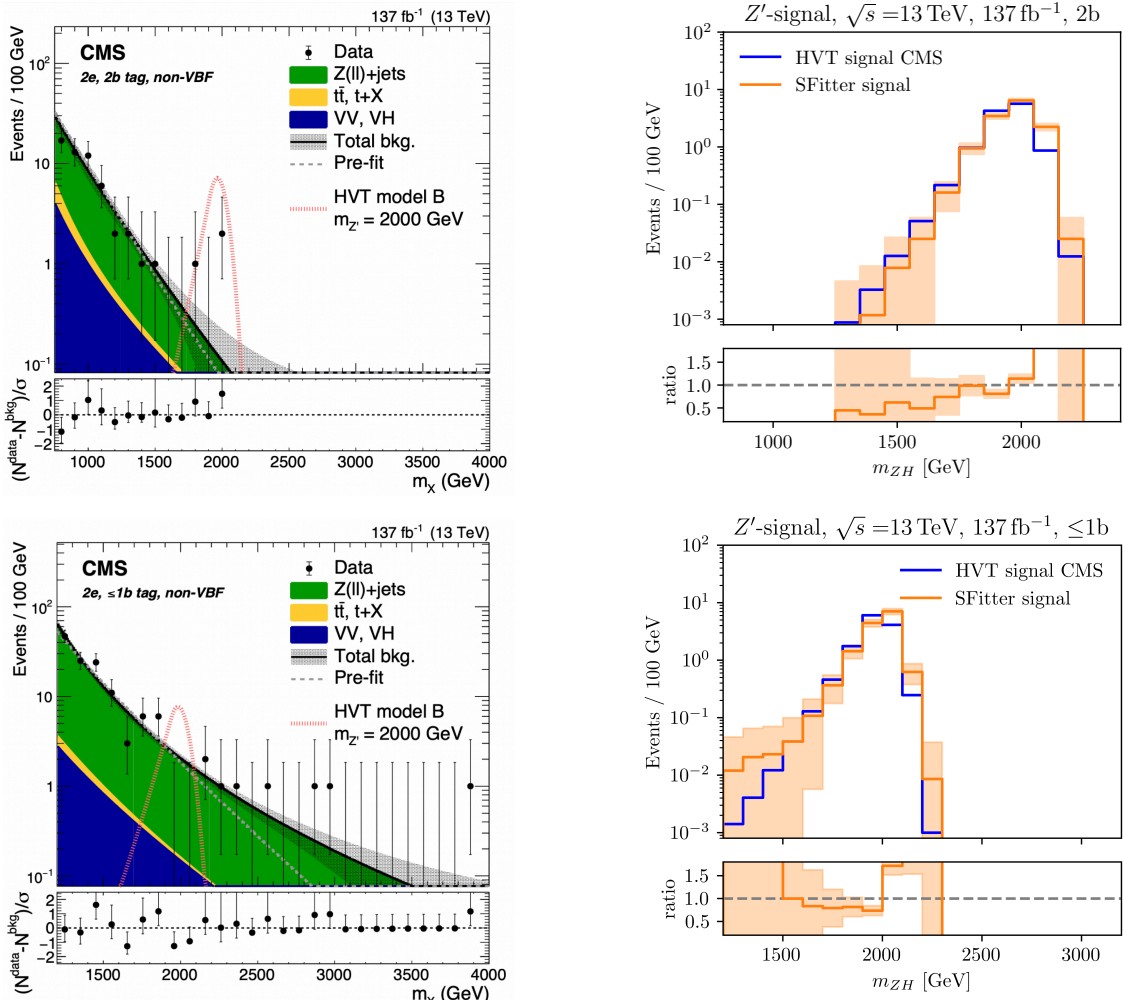

Figure 15: Left: measured $m_{ZH}$ distributions for the two $b$-tagging categories [79]. Right: comparison between the $Z'$ signal quoted by CMS and our estimate. The orange bands show the statistical uncertainty from the Monte Carlo generation.

This signal has the advantage that it is localized in $m_{ZH}$ and simulated at leading order using Madgraph, which means it is easier to use for calibration than a continuum background. Again, we use Madgraph, Pythia, FastJet, and Delphes with the standard CMS card at leading order. The combined sample is then compared to the HVT peak shown in Fig. 15. We extract the experimental efficiencies after scaling the invariant mass by the same factor 1.05 for both categories. The right panels in Fig. 15 show the simulated $Z'$ signal for the two categories, compared with the quoted CMS distributions.

The SMEFT signal in the $ZH$ channels is then computed using the same efficiencies and the reweighting module in Madgraph. The $Z$-decays are included in the Madgraph simulation, while the Higgs decays are simulated in Pythia. As before, we ignore SMEFT effects on the $t\bar{t}$ background.

Also for the CMS $ZH$ channel we need to re-bin the $m_{ZH}$ distribution to define a meaningful set of measurements, now with at least two events per bin and separately for the two categories. The results are shown in Fig. 16. For each bin we include the systematic and statistical uncertainties from Ref [79]. In addition, we include different theory uncertainties per bin from the SMEFT prediction and event generation in Madgraph. As discussed in detail for

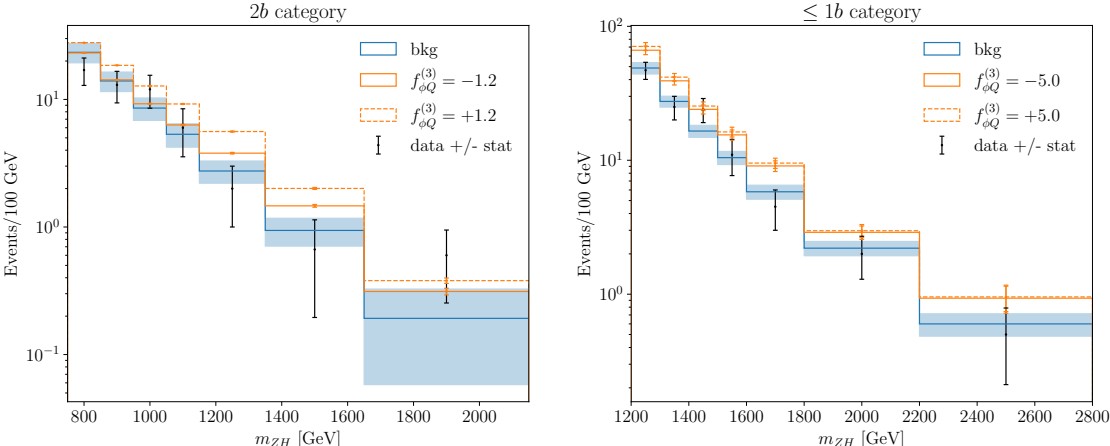

Figure 16: Re-binned $m_{ZH}$ distributions for the $2b$ category (left) and the $\leq 1b$ category implemented in SFitter, including statistical and systematic uncertainties. We show the complete continuum background and the effect of a finite Wilson coefficient $f_{\phi Q}^{(3)}$.

the ATLAS $WH$ analysis, we neglect the correlation between bins.

## 4.4  Boosted Higgs production

Boosted Higgs production, in association with one or more hard jets,

$$pp \rightarrow Hj(j)\,, \tag{30}$$

has been known to distinguish between a top-induced Higgs-gluon-gluon coupling and the corresponding dimension-6 operator for a long time [81, 82]. It has therefore been suggested as a channel to measure the dimension-6 Wilson coefficient $f_{GG}$ in the presence of a modified top Yukawa coupling $f_t$ [83–86], where it competes with channels like the off-shell Higgs production [87, 88]. In the SFitter Higgs analysis it can be added to the set of measurements to provide complementary information to the total Higgs production rate. We take the measurement of the Higgs $p_T$ distribution in the $\gamma\gamma$ channel by ATLAS [80].

The main contribution to boosted Higgs production comes from the partonic channel $gg \rightarrow Hg$, with subleading corrections from $gg \rightarrow Hgg$. This allows us to include SMEFT corrections to $gg \rightarrow Hg$ only. They can be separated into rescalings of the top Yukawa coupling, for instance via $\mathcal{O}_{u\phi,33}$, corrections to the top-gluon coupling from $\mathcal{O}_{tG}$, and the effective Higgs-gluon interaction induced by $\mathcal{O}_{GG}$.

Because these effective vertices enter also $t\bar{t}H$ production, these operators lead to a non-trivial interplay in the global analysis. Moreover, as discussed in Sec. 4.5 below, $f_{tG}$ is well-constrained by top pair production $pp \rightarrow t\bar{t}$. In fact, it constitutes the most significant contact between global top and Higgs analyses [12, 13].

We calibrate the boosted Higgs analysis simulating the SM signal for the partonic sub-channels $gg \rightarrow Hg$ and $gg \rightarrow Hgg$ using Madgraph. The gluon-initiated channels are simulated at 1-loop, while the quark-initiated one at tree level. For the one-loop simulation we use a fixed renormalization scale $\mu_R = m_H$. This setup is also used for the SMEFT simulations. Figure 17 shows the comparison between our simulation and the SM signal estimate provided by ATLAS. We use the same binning as in the original distribution, but omit the bins with $p_{T,\gamma\gamma} < 45$ GeV.

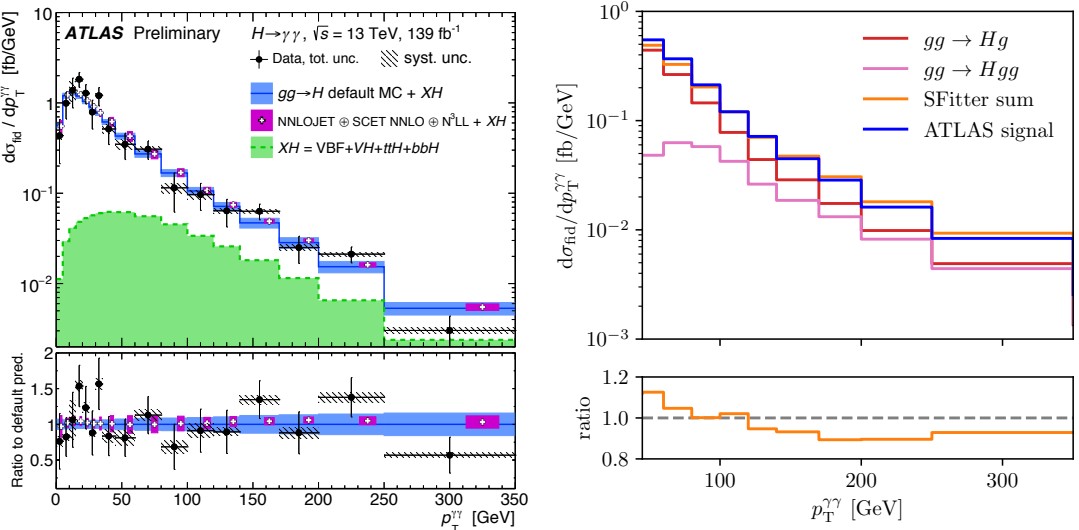

Figure 17: Left: measured $d\sigma_{\text{fid}}/dp_T^{\gamma\gamma}$ distribution [80]. Right: comparison between the the ATLAS distribution and our SM estimate summing contributions from $gg \rightarrow Hg$ and $gg \rightarrow Hgg$.

The simulation of SMEFT effects is tackled with different methods. The effect of a shifted top Yukawa is just a rescaling of the SM cross section, that can be easily computed analytically,

$$\frac{\sigma_{\text{SMEFT}}}{\sigma_{\text{SM}}} = \left(1 - \frac{f_t}{\sqrt{2}} \frac{v^2}{\Lambda^2}\right)^2 . \tag{31}$$

Second, $\mathcal{O}_{tG}$ also enters the top loops, but induces a different Lorentz structure compared to the SM amplitude. Its contributions are simulated independently using SMEFT@NLO [89] in Madgraph. In the event generation, the EFT operator is renormalized at $\mu_{\text{EFT}} = \mu_R = m_H$.

Finally, $\mathcal{O}_{GG}$ enters at the tree level. Because the pure interference between tree and loop diagrams cannot be generated directly in Madgraph, we choose to simulate both the linear and the squared term with a modified `loop_sm` UFO model, where the point-like Higgs-gluon vertices are mimicked by sending the bottom quark mass and Yukawa coupling to 15 TeV. We verified that any value larger than 10 TeV gives equivalent results. This way the simulation is formally at one loop for all terms. The results of this approximation were cross-checked against the analytic results in Refs. [81, 82] for the interference and against the tree-level simulation for the pure square.

The mixed quadratic terms, *i.e.* the interferences between two operators, can be computed analytically for the combination of $\mathcal{O}_{tG}$ or $\mathcal{O}_{GG}$ with a shifted Yukawa coupling. The combination of $f_{tG}$ and $f_{GG}$ needs to be simulated independently, in our case using using SMEFT@NLO and the reweighting module in Madgraph.

In Fig.18 we show the impact of four relevant SMEFT coefficients on the kinematic distribution we implement in SFitter. For each bin we include the systematic and statistical uncertainties from Ref [80], as well as an additional 20% theory uncertainty reflecting the scale uncertainty on the SMEFT prediction.

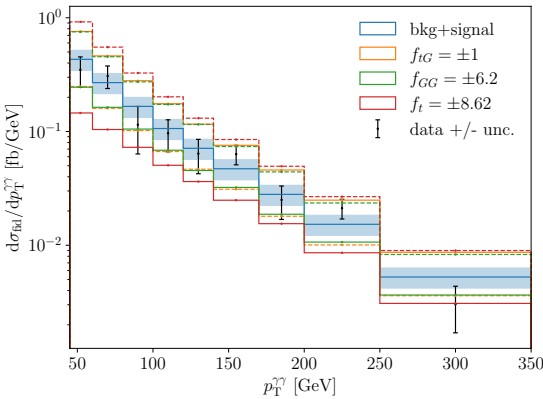

Figure 18: Reconstructed $p_{T,H}$ distribution implemented in SFitter, including statistical and systematic uncertainties as well as additional uncertainties on our prediction. We show the complete continuum of signal and background and the effect of three finite Wilson coefficients $f_t$, $f_{tG}$ and $f_{GG}$. The negative values are represented by dashed lines and the positive values by solid lines.

## 4.5 From the top

From the combined top-Higgs analyses [12, 13] we know that the Higgs-gauge sector and the top sector cannot be treated completely independently. The two operators

$$\mathcal{O}_{u\phi,33} = \phi^\dagger\phi\,\bar{Q}_3\tilde{\phi}u_{R,3} \qquad \text{and} \qquad \mathcal{O}_{tG} = ig_s(\bar{Q}_3\sigma^{\mu\nu}T^A u_{R,3})\,\tilde{\phi}G^A_{\mu\nu} \qquad (32)$$

contribute to top pair and associated $t\bar{t}H$ production and are, at the same time, crucial to interpret gluon-fusion Higgs production, together with the Higgs-related operator $\mathcal{O}_{GG}$, as discussed above. By the definition of top-sector and Higgs-sector SMEFT analyses in SFitter, $\mathcal{O}_{tG}$ is covered by the top analysis, while we keep $\mathcal{O}_{u\phi,33}$ as part of the Higgs analysis, together with a complete treatment of $t\bar{t}H$ production. This means we can include the limits on $f_{tG}$ from the dedicated SFitter analysis of the top sector [11] using its 1-dimensional profile likelihood. We implement these constraints as an external measurement or prior. The corresponding profile likelihood is shown in Fig. 19. It consists of 100 data points which are dense enough that we can linearly interpolate between them.

We choose the range in $f_{tG}$ to cover extremely small log-likelihoods, to avoid numerical issues in the combined analysis. Still, while it is very unlikely to occur, we also want to describe points outside of this range, so we extrapolate the log-likelihood further with two quadratic

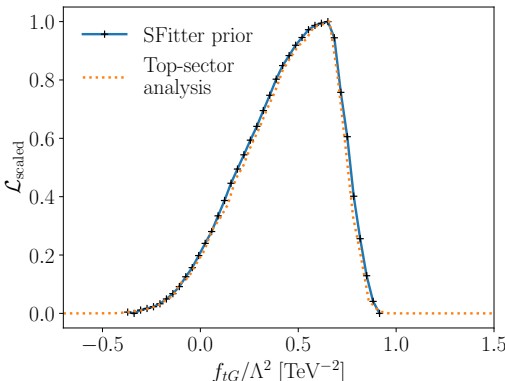

Figure 19: Profile likelihood for $f_{tG}$ from the SFitter top-sector analysis [11].

fits; one fitted to negative Wilson coefficients and one fitted to positive Wilson coefficient. A quadratic fit in this context means exponentially suppressed Gaussian tails.

## 4.6   Rates and signal strengths

In addition to the new kinematic measurements above, we update the set of Higgs rate measurements of Ref. [3], adding those listed in Tab. 1. The two $H \to \tau\tau$ and three out of four $H \to$ inv measurements are completely new constraints, while the others update results included in our previous analysis. The first column indicates which production channels were implemented in SFitter. We do not always use all the channels covered in a given ATLAS or CMS paper, if some of them are clearly subleading or some of them appear impossible to implement in the necessary details. Production channels in parentheses are numerically subleading, but were retained nevertheless.

The systematic and statistical uncertainties of the new measurements are typically smaller compared to the older ones. On the other hand, we attempt a more comprehensive and conservative estimate of the theory uncertainties, given the available information. In Ref. [3] we typically discarded many theory uncertainties on the signal quoted in the actual papers and replaced them with the leading uncertainty on the complete signal prediction from the HXSWG [101–103], added linearly as expected for uncorrelated flat uncertainties combined by profiling. In our new, comprehensive treatment, all theory uncertainties quoted by the analyses are retained. We include them separately and combine them. In addition, we include the uncertainties reported by the HXSWG [101–103] as the uncertainty on our SFitter prediction, again split by contribution and ready to be profiled over or marginalized.

We illustrate the implementation procedure in some more detail only for the recent Run-2 $H \to WW$ analysis by CMS [98]. Among the results presented, we implement the four signal strength measurements. Because they are reported for individual production modes (and not only in the STXS binning), they can be directly compared to the known expressions for Higgs production rates in the SMEFT, without re-deriving. These have been long implemented in SFitter for the main Higgs production channels (ggF, VBF, $WH$, $ZH$, $ttH$) and decays ($b\bar{b}$, $WW$, $gg$, $\tau\tau$, $ZZ$, $\gamma\gamma$, $Z\gamma$, $\mu\mu$). A re-derivation of the SMEFT expression can also be avoided in cases where the final results are not given for specific production channels, but the expected signal contribution from each production channel is provided.

The key ingredient to SFitter is a detailed breakdown of all uncertainties. This is crucial

| Production | Decay | ATLAS | CMS |
|---|---|---|---|
| All | $H \to \gamma\gamma$ | [90] | [91] |
| $ZH$ | $H \to$ inv | [92] | [93] |
| VBF (ggF, $VH$) | $H \to$ inv | [94] | |
| VBF (ggF, $ZH$, $t\bar{t}H$) | $H \to$ inv | | [95] |
| All | $H \to \tau\tau$ | [96] | |
| $VH$ | $H \to \tau\tau$ | | [97] |
| ggF, VBF | $H \to WW$ | [78] | |
| ggF, VBF, $VH$ | $H \to WW$ | | [98] |
| $WH, ZH$ | $H \to b\bar{b}$ | [99] | |
| ggF, VBF ($VH$, $ttH$) | $H \to \mu\mu$ | | [100] |

Table 1: List of the new Run 2 Higgs measurements included in this analysis, we denote $V = W, Z$.

in order to obtain the best possible approximation of the full experimental likelihood. For Ref. [98] we consider different uncertainties for each production channel, that are reported in the paper and in the corresponding HepData entry.

The statistical uncertainty is taken from the experimental paper, symmetrized and implemented as Poisson or Gaussian distribution. For experimental systematics, SFitter provides 31 predefined categories of Gaussian uncertainties, correlated across measurements and, where appropriate, across experiments. All uncertainties belonging to the same category are added in quadrature. The categories used to implement the CMS analysis cover luminosity, detector effects, lepton reconstruction, and b-tagging. Detector effects combine the jet energy scale and resolution uncertainties, as well as the missing transverse momentum scale uncertainty. Whenever the experimental papers quote significant uncertainties that do not fit any predefined category, we add them as an uncorrelated Gaussians, but this is not the case for the analysis of Ref. [98].

Theoretical uncertainties are typically implemented with flat uncorrelated likelihoods. One exception is the Monte Carlo statistics uncertainty, which we usually treat as an uncorrelated Gaussian. The CMS analysis quotes five theoretical uncertainties, that are all introduced independently. In addition, we have six theoretical uncertainties on the SFitter prediction: three on the production rate and three on the decay branching ratio, following the HXSWG prescription [101–103].

As a final step we compare the systematic uncertainties quoted on the final result with the sum of the uncertainties implemented in SFitter. If we are missing information for example on the correlations, our implementation might not be conservative, so we introduce an additional uncorrelated Gaussian uncertainty to compensate. This happens for the CMS reference analysis in the $ZH$ channel. For this measurement we implement two uncorrelated Gaussian uncertainties, three correlated Gaussian uncertainties, plus the eleven flat uncertainties.

# 5   Global SFitter analysis

After validating the marginalization technique in SFitter and introducing a set of promising new observables, we can provide the final global analysis of the Higgs and electroweak sector after Run 2, including the leading link to the top sector. To be conservative, we will compare all our results with a profile likelihood treatment. We will find and explain differences of the two methods facing the same extended dataset.

## 5.1   Marginalization vs profiling complications

While in Sec. 3 we have found that for the dataset of Ref. [3] the marginalization and profiling approaches lead to, essentially, identical results, one analysis implemented in SFitter as part of Ref. [64] actually leads to significant differences. The data driving this separation of profiling and marginalization is the $m_{WW}$ distribution measured by ATLAS [104], shown in the left panel of Fig. 11. It has the unique feature of a sizeable under-fluctuation in the last bin.

Such an under-fluctuation is challenging to accommodate in the SMEFT. First, under-fluctuations can only be explained by operators with large interference terms, where the Wilson coefficients have to be carefully tuned to be large enough to explain a sizeable effect and small enough to not be dominated by dimension-6 squared contributions. Second, a localized under-fluctuation in only one bin of one kinematic distribution requires a subtle balance of several Wilson coefficients, to control all other bins in all other di-boson and $VH$ channels.

In Fig. 20 we show low-dimensional analyses of the full Run 2 dataset including the $WW$ kinematics shown in Fig. 11, constraining three, five and seven Wilson coefficients. For the

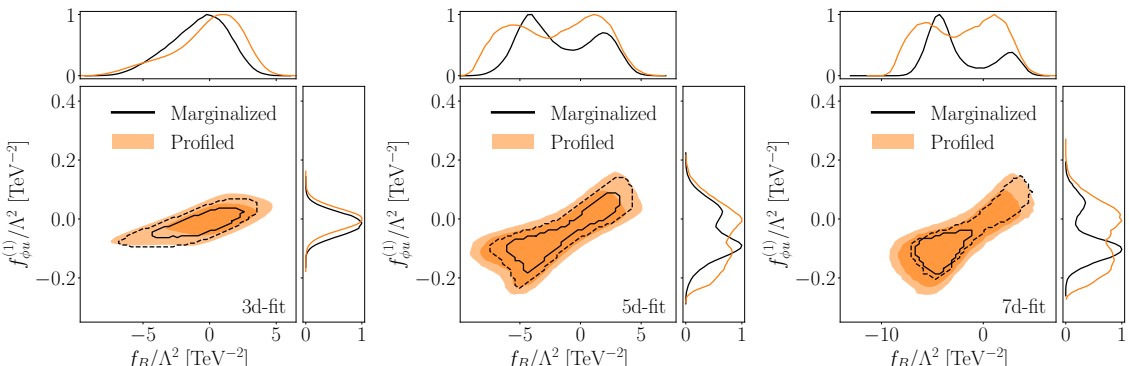

Figure 20: SFitter analysis with different SMEFT models describing the full Run 2 dataset, including the boosted $WW$ production.

three parameters $\{f_B, f_{\phi u}^{(1)}, f_W\}$ we see that the maximum of the likelihood is perfectly compatible with the SM. The reason is that the SMEFT model is not flexible enough to accommodate the under-fluctuation, so we only encounter the issue when we look at the value of the likelihood in the maximum. Adding first $\{f_{\phi Q}^{(1)}, f_{\phi Q}^{(3)}\}$ and then $\{f_{\phi d}^{(1)}, f_{3W}\}$ to the SMEFT model allows us to accommodate the under-fluctuation, leading to a second likelihood maximum.

When we compare the two likelihood maxima, differences between the profiling and the marginalization appear. This is not surprising, given that the two methods ask different questions. By definition, the profile likelihood identifies the most likely parameter point, which according to Fig. 20 is close to the SM point, $f_B \approx 0 \approx f_{\phi u}^{(1)}$. This does not change when we increase the operator basis or expressivity of the SMEFT model. The marginalization adds volume effects in the space of Wilson coefficients, and they increasingly prefer the non-SM maximum once the SMEFT model is flexible enough to explain the under-fluctuation. Consequently, the marginalized analysis proceeds to challenge the SM in favor of an alternative SMEFT parameter point.

## 5.2 Full analysis

After identifying and understanding the issue with marginalized likelihoods for the updated dataset we now perform the full, 21-dimensional parameters analysis on all available data. The theory framework is defined by the Lagrangian in Eq.(8). The dataset consists of all measurements from Ref. [3], combined with the new and updated channels described in Sec. 4. We will discuss the standard profile likelihood results below, in a first step we focus on the marginalization. In Fig. 21 we show a set of 1-dimensional marginalized likelihoods. In the first row we show three Wilson coefficients affected by the under-fluctuation in $m_{WW}$, as discussed in the previous Sec. 5.1. While the marginalized likelihood for $f_W$ follows a standard single-mode distribution, those for $f_B$ and $f_{\phi u}^{(1)}$, for example, show two distinct modes accommodating the observed under-fluctuation.

In the second row we show the alternative maximum in $f_+$ we already observed for the dataset from Ref. [3] and which we discuss in Fig. 6 of Sec. 3. For the final SFitter result we will remove the second maximum as an expansion around the wrong SMEFT limit. We also see that the invisible Higgs width is strongly constrained, even after we account for a modified Higgs production process rather than assuming SM Higgs production combined with the exotic invisible Higgs decay.

In the last row we show the effect of including $\mathcal{O}_{tG}$ in the Higgs analysis. Comparing the limit on $f_{tG}$ to its prior in Fig. 19 we see that this parameter gains essentially nothing from the

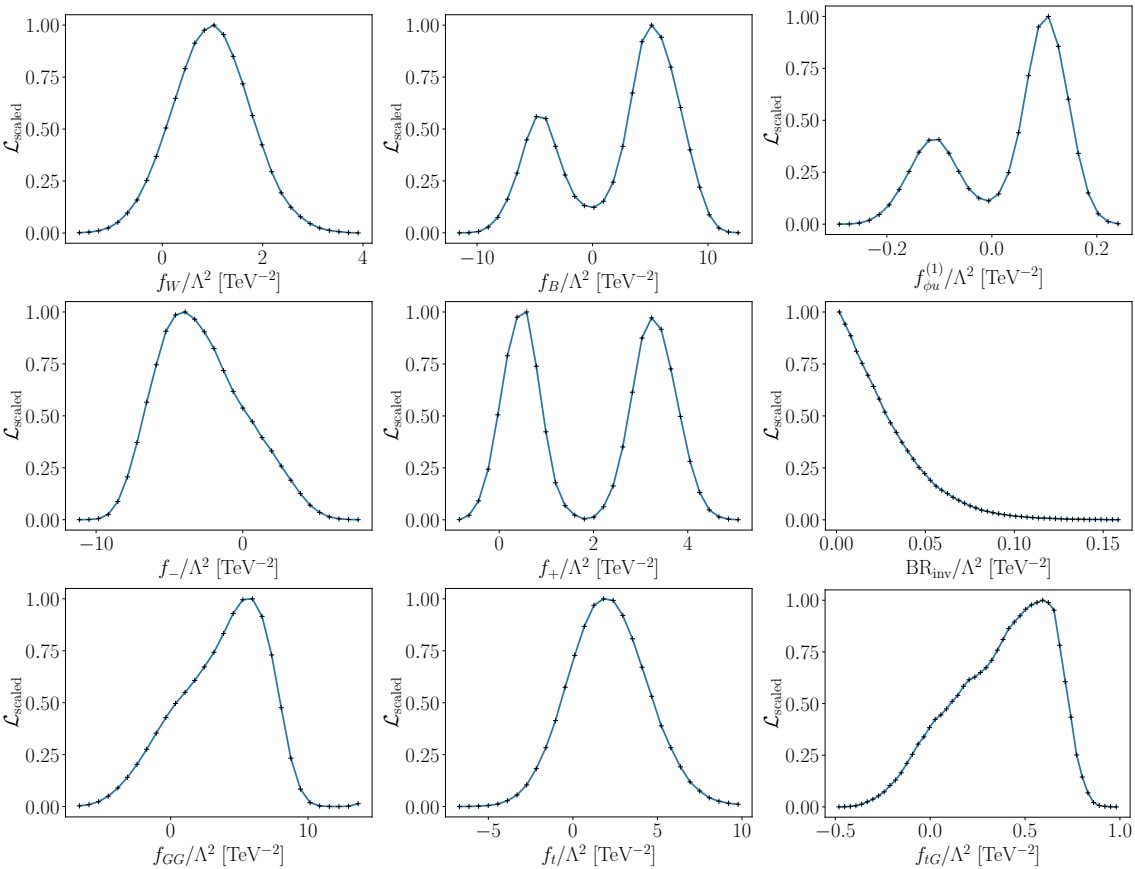

Figure 21: Set of marginalized likelihoods for the 21-dimensional SFitter analysis including the full set of measurements.

Higgs measurements, but it will broaden the limits on the correlated parameter $f_{GG}$ affecting gluon-fusion Higgs production.

To follow up on the discussion of Fig. 20 we show a more complete set of 2-dimensional marginalized likelihoods related to the $m_{WW}$ under-fluctuation in Fig. 22. In the full analysis the correlation does not just affect $f_{\phi u}^{(1)}$, but the full range of gauge-fermion operators. This

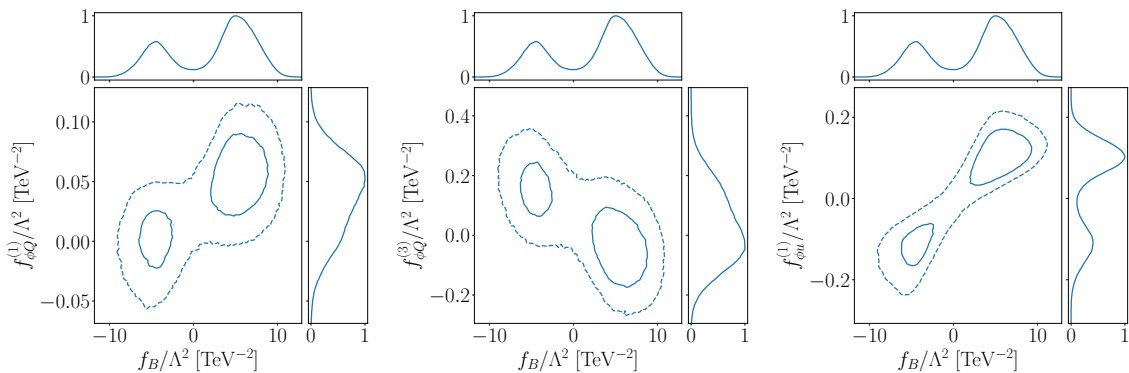

Figure 22: Set of marginalized correlations for the 21-dimensional SFitter analysis including the full set of measurements. The solid and dashed lines show $\Delta \chi^2 = 2$ and 7 respectively.

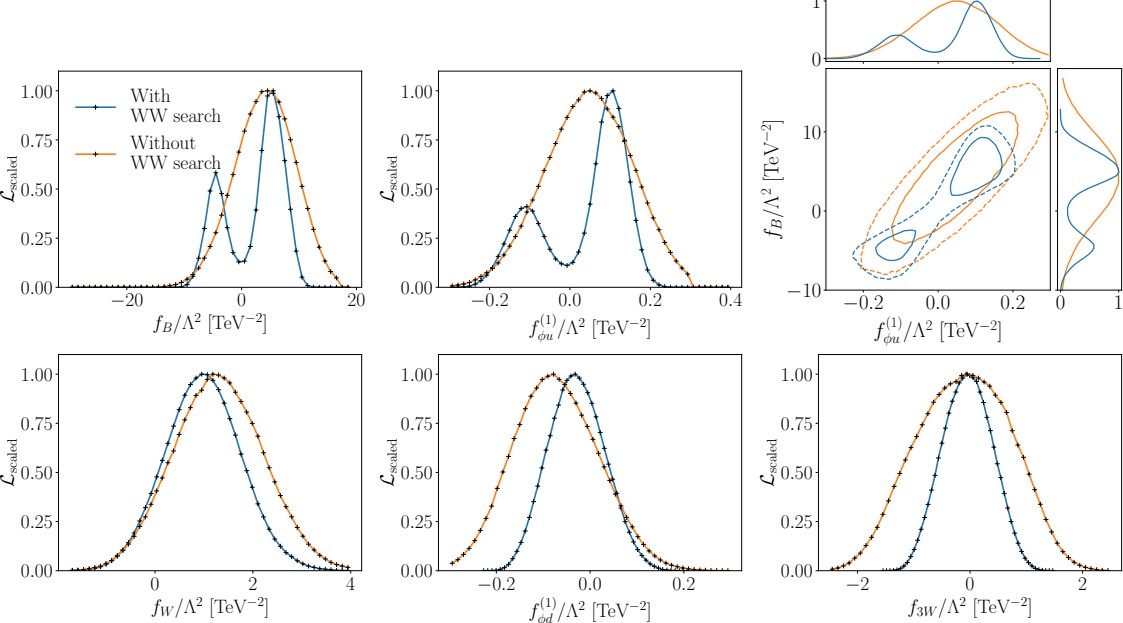

Figure 23: Set of marginalized likelihoods for the 21-dimensional SFitter analysis with and without the ATLAS $WW$ resonance search altogether.

is expected from the argument that we need to carefully tune many Wilson coefficients to accommodate a deviation in a single di-boson process in a single bin of the high-invariant-mass distribution. As mentioned before, the apparent signal for physics beyond the Standard Model is an artifact of the marginalization and its volume effects, and cannot be reproduced with the profile likelihood. Note that this does not mean the marginalization is wrong or wrongly done, this difference just reflects the two methods asking different questions.

To study the impact of the critical $WW$-resonance analysis on our global analysis we show a set of marginalized likelihoods with and without this analysis, *i.e.* with and without the entire $m_{WW}$ distribution. Obviously, removing this distribution also removes the secondary maximum structure, as we immediately see in Fig. 23. Removing the entire distribution replaces the marginalized likelihoods for $f_B$ and $f_{\phi u}^{(1)}$ by their broad envelopes, still correlated, but without the distinctive maxima. For $f_W$ the additional observable has limited impact, for $f_{\phi d}^{(1)}$ is leads to a smaller uncertainties combined with a shifted maximum, and for $f_{3W}$ the $WW$-analysis provides key information.

Finally, in Fig. 24 we compare the 1-dimensional marginalized likelihoods with the cor-

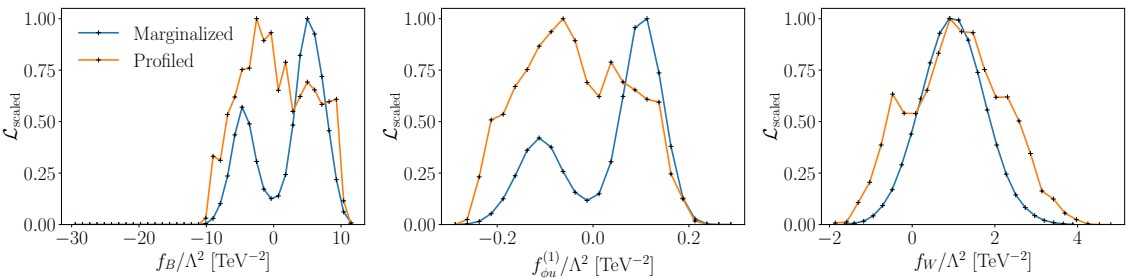

Figure 24: Set of marginalized and profiled likelihoods for the 21-dimensional SFitter analysis with the ATLAS $WW$ resonance search.

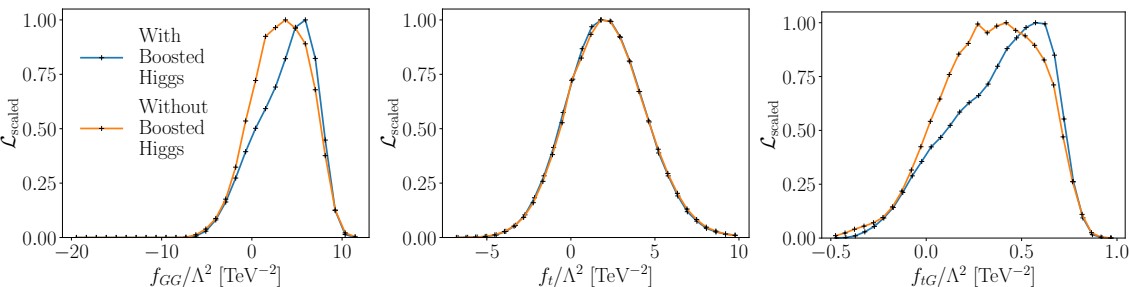

Figure 25: Set of marginalized likelihoods for the 21-dimensional SFitter analysis with and without the boosted Higgs analysis.

responding profile likelihoods for a set of Wilson coefficients. For $f_B$ and $f_{\phi u}^{(1)}$ we see the difference in the treatment of the secondary likelihood maximum, while $f_W$ serves as an example for the many parameters where the two methods give the same results, as discussed in detail in Sec. 3 and Fig. 2. Indeed, the results from the two methods only disagree when the likelihoods develop secondary maxima.

Moving on with the effects observed in Fig. 21 we can look at the top-Higgs sector with $f_{GG}$, $f_t$, and the added $f_{tG}$. These three Wilson coefficients are constrained by the Higgs production in gluon fusion, associated top-Higgs production, and top pair production through the prior shown in Fig. 19. We have already seen that this prior is practically identical to the final outcome in Fig. 21. Nevertheless, we can ask what the impact of the boosted Higgs production process is, given that it should provide a second measurement of the three Wilson coefficients with different relative weights. In Fig. 25 we show the results of the 21-dimensional SFitter analysis with and without the new boosted Higgs measurement introduced in Sec. 4.4. Unfortunately, the likelihood distributions are similar, corresponding to our expectation from the limited statistics of this measurements and the limited range in $p_{T,H}$, where significant differences can only be expected for $p_{T,H} > 250$ GeV [87], and even for this kinematic range it is not clear how well the measurement separates effects from $f_{GG}$ and $f_{tG}$, while the $f_t$ measurement is completely dominated by $t\bar{t}H$ production.

Even though completely justified, the only visible effect of including $f_{tG}$ in the Higgs analysis is to wash out the limit on $f_{GG}$. In Fig. 26 we first show the change on the 1-dimensional marginalized likelihood of $f_{GG}$ when we remove $f_{tG}$ from the SFitter analysis. Indeed, the measurement of $f_{GG}$ becomes much better. This is explained by the strong correlation between $f_{GG}$

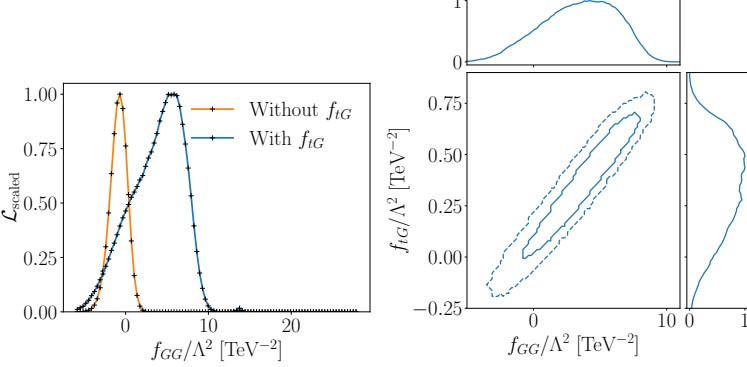

Figure 26: Left: marginalized likelihoods for the SFitter analysis with and without $f_{tG}$, using the same dataset; Right: marginalized correlation for the 21-dimensional SFitter analysis.

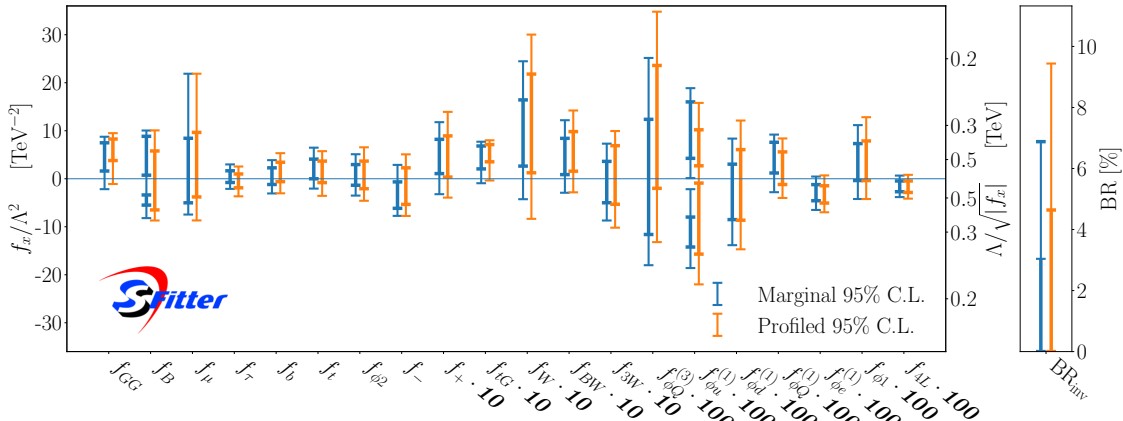

Figure 27: Comparison of 21-dimensional SFitter analysis with all updated measurements included. We show the 68% and 95%CL error bars from consistent marginalization and profile likelihood treatments of all nuisance parameters and Wilson coefficients.

and $f_{tG}$ shown in the right panel.

After the in-depth discussion of all features we show the 68% and 95%CL limits from the 21-dimensional SFitter analysis with the full updated dataset in Fig. 27. To extract these limits we start with the respective 1-dimensional marginal or profile likelihood, identify the maximum, and move outward keeping the likelihood values on the left and the right border of the integral the same. If there exists an additional peak, we compute the integral under the likelihood for the part of the curve above a given likelihood threshold. The 68% and 95%CL error bars are then defined the same way for the marginal and profile likelihood.

The profile likelihood results in Fig. 27 provide an update of the limits shown in Fig. 2 [3]. We emphasize that this update does not automatically mean an improvement of the limits, because of our more comprehensive uncertainty treatment, the added operator $\mathcal{O}_{tG}$, and the now measured Yukawa coupling $f_\mu$. Computing the uncertainties on the Wilson coefficients which are all in agreement with the Standard Model at least for the profile likelihood approach, we remove modes around non-SM likelihood maxima. Those appear through sign flips in Yukawa couplings and in $f_+$ and would require order-one effects from new physics. We safely assume that new physics with this kind of effects would have been observed somewhere already.

In Fig. 27 we see that all results from the marginalization and profiling approach are consistent with each other. The only kind-of-significant deviation appears in $f_B$ and the correlated gauge-fermion operators like $f_{\phi u}^{(1)}$. The reason for this discrepancy can be traced back to an under-fluctuation in the $m_{WW}$ measurement and actual differences between the likelihood and Bayesian approaches.

## 6   Outlook

Global SMEFT analyses are the first step into the direction of interpreting all LHC data on hard scattering process in a common framework. They allow us to combine rate and kinematic measurements from the Higgs-gauge sector, the top sector, jet production, exotics searches, even including parton densities and flavor physics. They can be considered improved binwise analyses of LHC measurements, but with a consistent effective theory framework. This framework allows us to provide precision predictions matching the precision of the data we

analyze, and it ensures that their result is relevant fundamental physics. Because any realistic effective theory description involves a truncation in dimensionality, SMEFT results always have to be considered in relation to the fundamental physics models they represent.

From a brief look at the analyzed data we know that our SMEFT analysis of the electroweak gauge and Higgs sector will not describe established anomalies, but serve as a consistent, global limit-setting tool. This makes it even more important to treat all uncertainties, statistical, systematic, and theory, completely and consistently. Technically, this leads us directly to the question if we want to use a profile likelihood or a Bayesian marginalization treatment. Because the two methods ask different questions, it is not at all clear that technically correct analyses following the two approaches lead to the same results. We have shown, for a first time, what the current challenges in global LHC analyses are and how the two methods do turn up slight differences.

We have started with an in-depth discussion of the current challenges in the Higgs and electroweak data and the corresponding validation of the marginalization in SFitter, in comparison to our classic profile likelihoods. Using the established dataset of Ref. [3] we have shown that the two methods give extremely similar results. We have also found that for this dataset the exact treatment of the theory uncertainties is not a leading problem, while a correct treatment of correlations of the measurements and the uncertainties is crucial.

Next, we have updated this dataset, including a set of kinematic di-boson measurements and boosted Higgs production. These measurements allow us to constrain operators with a modified Lorentz structure especially well. Kinematic distributions from di-boson resonance searches probe the largest momentum transfers of our SFitter dataset, but their interpretation in terms of SMEFT operators requires significant effort. A systematic publication of the corresponding likelihood by ATLAS and CMS would fundamentally change the appreciation for these analyses, from failed resonance searches to the most exciting SMEFT results.

Accidentally, the updated dataset also leads to differences in the marginalization and profiling treatments of the same exclusive likelihood. The measurement driving this difference is an under-fluctuation in the tail of the kinematic $m_{WW}$ distribution. Under-fluctuations are difficult to reconcile with SMEFT analyses, because they require a balance between linear and squared operator contributions. To complicate things, a sizeable number of kinematic distributions probes large momentum transfer, all consistent with the Standard Model. For a small number of Wilson coefficients one under-fluctuation will just lead to a poor log-likelihood value in the SM-like likelihood maximum. A larger number of Wilson coefficients defines a powerful model which accommodated this deviation. For the final result, the complex correlations between Wilson coefficients lead to volume effects in the marginalization, which, expectedly, separated the final profile likelihood and marginalized results.

## Acknowledgments

We would like to thank Anke Biekötter, Anja Butter and Tyler Corbett for ongoing help with SFitter details. For advice on Bayesian methods we thank Kevin Kröninger; for details on the different analyses we are grateful to Ines Ochoa, Pascal Baertschi, Alberto Zucchetta, and Christian Sander; for the implementation of $f_{tG}$ limits we had valuable support from Ken Mimasu and Eleni Vryonidou, and our Delphes questions were answered by Michele Selvaggi. Finally, we are grateful to Wolfgang Kilian and Michael Krämer for many discussions and to Kevin Gauss for his contributions to an early phase of the project. IB acknowledges funding from the Swiss National Science Foundation (SNF) through the PRIMA grant no. 201508. EG is supported by the International Max Planck School *Precision Tests of Fundamental Symmetries*. The research of all authors is supported by the Deutsche Forschungsgemeinschaft

(DFG, German Research Foundation) under grant 396021762 – TRR 257 *Particle Physics Phenomenology after the Higgs Discovery*. Last, but not least, we acknowledge support by the state of Baden-Württemberg through bwHPC and the German Research Foundation (DFG) through grant no INST 39/963-1 FUGG (bwForCluster NEMO).

## A    High luminosity LHC

Figure 28 shows the projected limits obtained with a marginalized treatment for the high-luminosity LHC on the 21 SMEFT Wilson coefficients presented in Eq. 8. We use the full set of measurements presented in Sec. 5, where all the LHC measurements were set to their background values and their luminosity scaled to $4\,\mathrm{ab}^{-1}$. The electroweak precision data from LEP is kept the same as in previous fits. We also derived the projected high-luminosity limits assuming halved theory and systematic uncertainties.

A significant gain is expected on the invisible branching ratio $\mathrm{BR}_{\mathrm{inv}}$ as well as the Yukawa corrections $f_t$, $f_b$ and $f_\tau$. The constraints improve for most of the other Wilson coefficients, with a few exceptions. Limits on $f_{tG}$ and $f_{GG}$ do not change because the main constraint on these operators stems from the external measurement or prior described in Sec. 4.5. As this constraint is the result of a global fit to several top processes, it does not simply scale as a statistical uncertainty and we did not change it. $f_{\phi e}^{(1)}$, $f_{\phi 1}$, $f_{4L}$ and $f_{BW}$ also exhibit stable constraints, because they are mostly set by the electroweak precision data. Both the top constraints and the electroweak precision data are also the reason why the constraints are not always centred around zero. Finally, the only operator for which the constraints worsen is $f_{3W}$, mostly due to the change in the central value. Indeed, this particular operator is especially sensitive to high kinematic searches for $VV$. Thus, the pure background assumption may lead to a reduced constraining power, as there are no under-fluctuations in the higher bins of the distributions.

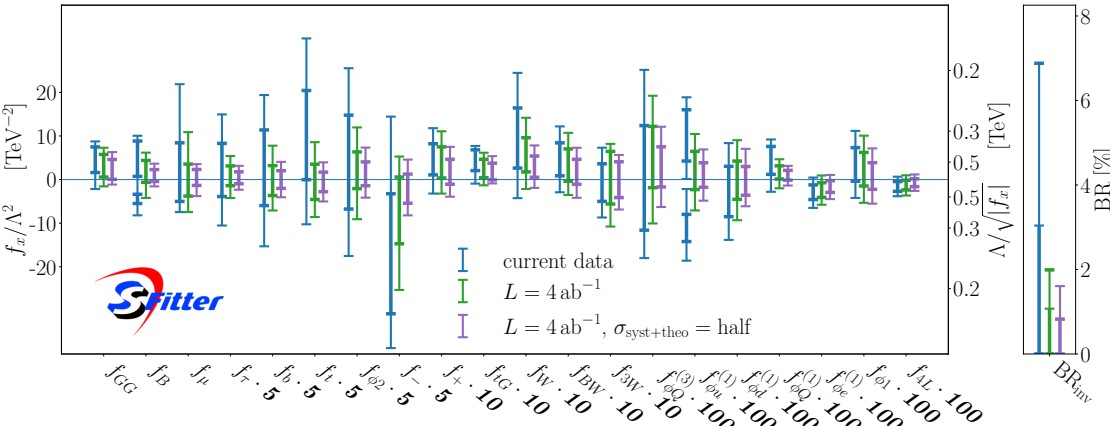

Figure 28: Green: 21-dimensional SFitter analysis with all updated measurements included, where all LHC measurements were scaled to a high-luminosity of $4\,\mathrm{ab}^{-1}$ and set to their background values. The electroweak precision data from LEP is kept the same as in previous fits. Purple: a second set of limits is derived setting all systematic and theoretical uncertainties to half their current values. We show the 68% and 95%CL error bars from a marginalization treatment of all the nuisance parameters and Wilson coefficients. Blue: current marginalized limits already presented in Fig. 27.

Once we also consider improved systematic and theoretical uncertainties, the limits on all Wilson coefficients improve—sometimes significantly, as for $f_\mu$ or $f_-$. In the case of $f_\mu$, this can be understood considering that the one analysis constraining this coefficient has a significance of only 3.0 standard deviations and is dominated by large experimental uncertainties.

## B  Numerical results

Table 2 reports the numerical values of the boundaries of the 68% and 95% CL intervals shown in Fig. 27.

| Coefficient | Marginalised | | Profiled | |
|---|---|---|---|---|
| | 68% CL | 95% CL | 68% CL | 95% CL |
| $f_{GG}$ | [1.61, 7.49] | [-2.17, 8.75] | [3.79, 8.28] | [-1.09, 9.50] |
| $f_B$ | [-5.49, -3.38] [0.74, 8,84] | [-8.20, 10.05] | [-6.49, 5.79] | [-8.69, 10.08] |
| $f_{\phi 2}$ | [-1.35, 2.95] | [-3.51, 5.11] | [-2.07, 3.68] | [-4.59, 6.55] |
| $f_\mu$ | [-5.01, 8.43] | [-7.45, 21.88] | [-3.79, 9.66] | [-8.68, 21.88] |
| $f_t$ | [-0.01, 4.09] | [-2.05, 6.47] | [-0.80, 3.68] | [-3.56, 5.75] |
| $f_b$ | [-1.19, 2.28] | [-3.06, 3.88] | [-0.60, 3.44] | [-3.03, 5.33] |
| $f_\tau$ | [-0.78, 1.66] | [-2.11, 2.99] | [-1.88, 1.00] | [-3.66, 2.55] |
| $f_-$ | [-6.16, -0.65] | [-7.73, 2.89] | [-5.34, 2.28] | [-7.75, 5.09] |
| $f_+ \times 10$ | [1.07, 8.21] | [-3.21, 11.79] | [0.36, 8.93] | [-3.93, 13.93] |
| $f_{tG} \times 10$ | [2.05, 6.82] | [-0.93, 7.72] | [3.53, 7.12] | [-0.36, 8.02] |
| $f_W \times 10$ | [2.64, 16.43] | [-4.26, 24.47] | [1.25, 21.80] | [-8.35, 30.03] |
| $f_{BW} \times 10$ | [0.86, 8.42] | [-2.91, 12.19] | [1.57, 9.82] | [-2.83, 14.22] |
| $f_{3W} \times 10$ | [-5.0, 3.62] | [-8.7, 7.31] | [-5.31, 6.89] | [-10.19, 9.94] |
| $f_{\phi Q}^{(3)} \times 100$ | [-11.61, 12.38] | [-18.01, 25.17] | [-1.99, 23.60] | [-13.19, 34.80] |
| $f_{\phi u}^{(1)} \times 100$ | [-14.22, -7.98] [4.25, 16.01] | [-18.60, -2.17] [0.14 18.87] | [-15.70, -0.90] [2.70, 10.20] | [-22.00, 15.80] |
| $f_{\phi d}^{(1)} \times 100$ | [-8.51, 3.04] | [-13.84, 8.37] | [-8.64, 6.07] | [-14.69, 12.13] |
| $f_{\phi 1} \times 100$ | [-0.37, 7.32] | [-4.22, 11.17] | [-0.37, 7.88] | [-4.22, 12.83] |
| $f_{4L} \times 100$ | [-2.7, -0.46] | [-3.82, 0.66] | [-2.86, -0.46] | [-4.14, 0.82] |
| $f_{\phi Q}^{(1)} \times 100$ | [1.2, 7.6] | [-2.8, 9.2] | [-1.20, 5.60] | [-4.00, 8.40] |
| $f_{\phi e}^{(1)} \times 100$ | [-4.58, -1.22] | [-6.5, 0.46] | [-5.06, -1.46] | [-6.98, 0.67] |
| $BR_{inv}$ | [0, 3.04] | [0, 6.88] | [0, 4.64] | [0, 9.44] |

Table 2: Numerical values for the results shown in Fig. 27.

Table 3 report the numerical values of the boundaries of the 68% and 95% CL intervals shown in Fig. 28.

| Coefficient | $L = 4\,\text{ab}^{-1}$ | | $L = 4\,\text{ab}^{-1}, \sigma_{\text{syst+theo}} = \text{half}$ | |
|:---:|:---:|:---:|:---:|:---:|
| | 68% CL | 95% CL | 68% CL | 95% CL |
| $f_{GG}$ | [0.42, 5.45] | [-1.78, 7.02] | [-0.18, 4.03] | [-1.42, 6.01] |
| $f_B$ | [-0.41, 3.68] | [-3.04, 6.02] | [-0.21, 2.2] | [-1.74, 3.3] |
| $f_{\phi 2} \times 5$ | [-2.85, 4.64] | [-9.09, 9.63] | [-1.17, 3.89] | [-4.2, 6.42] |
| $f_\mu$ | [-5.01, 5.99] | [-9.9, 21.88] | [-1.34, 2.32] | [-3.79, 5.99] |
| $f_t \times 5$ | [-4.77, 3.74] | [-9.63, 8.61] | [-3.09, 2.17] | [-5.72, 4.8] |
| $f_b \times 5$ | [-3.96, 2.57] | [-9.19, 7.8] | [-1.91, 2.11] | [-4.32, 4.12] |
| $f_\tau \times 5$ | [-1.45, 3.3] | [-6.2, 7.1] | [-1.01, 1.88] | [-2.75, 3.03] |
| $f_- \times 5$ | [-14.31, 0.84] | [-22.39, 5.89] | [-4.88, 1.47] | [-8.06, 4.12] |
| $f_+ \times 10$ | [0.36, 7.5] | [-3.93, 11.07] | [-1.07, 3.93] | [-3.93, 6.79] |
| $f_{tG} \times 10$ | [-0.13, 4.13] | [-1.46, 6.0] | [-0.14, 3.39] | [-1.18, 5.06] |
| $f_W \times 10$ | [0.59, 7.28] | [-2.75, 12.63] | [0.29, 4.55] | [-2.03, 6.48] |
| $f_{BW} \times 10$ | [-0.05, 7.15] | [-3.42, 11.0] | [-1.5, 3.88] | [-3.98, 7.19] |
| $f_{3W} \times 10$ | [-4.44, 5.17] | [-8.92, 7.09] | [-3.03, 3.29] | [-5.35, 4.62] |
| $f_{\phi Q}^{(3)} \times 100$ | [-0.9, 11.55] | [-8.37, 19.01] | [-1.44, 7.22] | [-5.78, 11.55] |
| $f_{\phi u}^{(1)} \times 100$ | [-1.45, 6.44] | [-6.3, 8.86] | [-1.63, 3.64] | [-4.5, 6.51] |
| $f_{\phi d}^{(1)} \times 100$ | [-4.21, 3.46] | [-8.4, 7.65] | [-2.89, 2.97] | [-5.6, 6.13] |
| $f_{\phi 1} \times 100$ | [-1.1, 6.44] | [-4.88, 11.3] | [-2.51, 3.29] | [-5.41, 6.67] |
| $f_{4L} \times 100$ | [-2.37, 0.02] | [-3.65, 1.14] | [-1.46, 0.21] | [-2.43, 1.04] |
| $f_{\phi Q}^{(1)} \times 100$ | [-0.21, 2.65] | [-2.11, 4.32] | [-0.32, 1.74] | [-1.45, 2.86] |
| $f_{\phi e}^{(1)} \times 100$ | [-4.23, -0.69] | [-5.89, 1.44] | [-2.68, -0.24] | [-4.1, 0.98] |
| $\text{BR}_{\text{inv}}$ | [0, 1.05] | [0, 2.26] | [0, 0.85] | [0, 1.75] |

Table 3: Numerical values for the results shown in Fig. 28.

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
