# Peer review of "To Profile or To Marginalise -- A SMEFT Case Study"

_SciPost Physics_

## Round 1 · Referee Report · Anonymous (Referee 1) · 2023-4-17

Report

Dear Editors,

In the manuscript To Profile or To Marginalize - A SMEFT Case Study - SciPost submission 202212_00023v1, the authors study, in the context of global SMEFT analyses, the results coming from profile likelihoods and Bayesian marginalisation frameworks.

After introducing the SMEFT operator basis they use, they present an overview of the SFitter framework, focusing on the marginalisation procedure, where they obtain the marginal likelihood of a measurement by integrating the fully exclusive likelihood over a set of nuisance parameters. They also comment on their treatment of uncertainties and correlations, the combination of channels and measurements, and the validation of their methodology. A first comparison between the two methods, profile likelihood and marginalisation, is shown by validating against an older study and dataset. The results show an overall good agreement and relevant differences are properly highlighted.

Afterwards, they go over the updated dataset they use for their study which includes diboson, Higgs, and top measurements. They comment on the differences and complications that can arise when comparing profile/marginalisation approaches and, in particular, the influence of the ATLAS $WW$ invariant mass distribution with an under-fluctuation in the last bin. The authors finally present a full analysis of likelihoods and correlations in their 21-dimensional study of Wilson coefficients.

The work presented in this article is of high quality, and of particular relevance in the context of indirect searches for New Physics at the LHC. In particular, it illustrates the potential differences that can arise when using profile likelihoods and Bayesian marginalisation in the SMEFT framework.
The manuscript is well presented and conveys a clear, self-contained message. I can, without hesitation, recommend it for publication as a regular article in SciPost. I list below some minor comments that the authors may consider addressing in their final version of the manuscript.

  1. The use of British and American spellings of words such as marginalisation/marginalization or marginalised/marginalized are both present in the manuscript. The authors might consider sticking to one of the two.

    1. In the introduction of the manuscript the acronym EFT is introduced but never explicitly defined, as done for SM and SMEFT.

    2. In the introduction, when mentioning studies of EFTs with parton densities, the authors might consider adding to Ref. 14 the papers 1905.05215, 2104.02723 and 2303.06159. The last one has been released after the submission of the present work, but the authors might consider adding it as well.

    3. At the beginning of section 3, in point 1., the authors claim that established anomalies should be discussed using properly defined BSM models. Could the authors clarify what they mean by this? Often $B$-anomalies, $g-2$, $m_W$, etc. anomalies are interpreted within an EFT framework.

    4. In Fig. 1, the authors say that the orange curve is showing the Gaussian obtained by summing in quadrature the half widths. Why is it that the orange curve in the right figure is narrower than the one on the left?

    5. In Figs. 3 and 4, a comparison between the marginalised and profile likelihood is shown. Why is the profile likelihood result so noisy? Can the authors comment on it in the text?

    6. At the beginning of page 11, referring to Fig. 2, the authors say that the figure displays excellent agreement between Bayesian marginalisation and profile likelihood. However, some coefficients show non negligible differences. For instance, $f_B$ shows a roughly $25\%$ shift, $f^{(3)}_{\phi Q}$ has a $15\%$ difference, as well as $f^{(1)}_{\phi u}$. Maybe the authors could consider commenting on it.

    7. In Sec. 3, in the text related to Fig. 6, the authors discuss the presence of a second mode in $f_+$ and how they tested its importance in the bounds of other Wilson coefficients. In particular, they run a Markov chain to map out the posterior distribution and then separate the samples. Could this be done more efficiently within the SFitter framework by simply applying a prior on $f_+$?

    8. At the end of Sec. 5.1, the authors comment on the difference between the marginalised and profile likelihood saying that the two methods ask different questions. I suggest that the authors elaborate more on that and make clearer which questions are asked and answered.

  • validity: -
  • significance: -
  • originality: -
  • clarity: -
  • formatting: -
  • grammar: -

Author:  Nina Elmer  on 2023-08-22  [id 3913]

(in reply to Report 1 on 2023-04-17)

We thank the referee for the careful revision of our manuscript. Here, we address the various
points raised in their reports and we explain how the manuscript has been modified.

1.The use of British and American spellings of words such as marginalisation/marginalization or marginalised/marginalized are both present in the manuscript. The authors might consider sticking to one of the two.

-> We changed the spelling and decided to stick to the American version

2.In the introduction of the manuscript the acronym EFT is introduced but never explicitly defined, as done for SM and SMEFT.

-> The acronym EFT is now defined in the introduction

3.In the introduction, when mentioning studies of EFTs with parton densities, the authors might consider adding to Ref. 14 the papers 1905.05215, 2104.02723 and 2303.06159. The last one has been released after the submission of the present work, but the authors might consider adding it as well.

-> We added the papers 1905.05215, 2104.02723 and 2303.06159

4.At the beginning of section 3, in point 1., the authors claim that established anomalies should be discussed using properly defined BSM models. Could the authors clarify what they mean by this? Often B
-anomalies, g−2, mW, etc. anomalies are interpreted within an EFT framework.

-> Anomalies in LHC physics, also those which we are looking for using global SMEFT analyses, usually affect a well-defined kinematic regime. For instance, enhanced tails in SMEFT analysis arise from tails of on-shell mass peaks far enough from the resonance, such that the EFT picture is applicable. Interpreting a sizeable kinematic anomaly naturally leads to an interpretation which makes these kinematic features explicit.

5.In Fig. 1, the authors say that the orange curve is showing the Gaussian obtained by summing in quadrature the half widths. Why is it that the orange curve in the right figure is narrower than the one on the left?

->sigma_{flat,tot} indicates in both panels the total uncertainty. For the right panel 3 flat uncertainties were used while in the left panel only one which is why the total uncertainty defined as the quadratic sum of the 'half-widths' is not the same. We added this explicitly to the caption of Fig. 1.

6.In Figs. 3 and 4, a comparison between the marginalised and profile likelihood is shown. Why is the profile likelihood result so noisy? Can the authors comment on it in the text?

-> We added a comment in the submission on page 10, the main reason is the way in which the profile likelihood is constructed that makes it noisy.

7.At the beginning of page 11, referring to Fig. 2, the authors say that the figure displays excellent agreement between Bayesian marginalisation and profile likelihood. However, some coefficients show non negligible differences. For instance, fB shows a roughly 25% shift, f(3)ϕQ has a 15% difference, as well as f(1)ϕu. Maybe the authors could consider commenting on it.

-> The occurring shifts are explained by the 2D-correlation plots in Fig. 5, additionally we changed the term 'excellent agreement' to 'good agreement'.

8.In Sec. 3, in the text related to Fig. 6, the authors discuss the presence of a second mode in f+
and how they tested its importance in the bounds of other Wilson coefficients. In particular, they run a Markov chain to map out the posterior distribution and then separate the samples. Could this be done more efficiently within the SFitter framework by simply applying a prior on f+?

-> This could be done by putting for instance flat priors over f+ which exclude one of the two modes. However, we also wanted to investigate what happens if we take both modes at the same moment into account, because otherwise some information on the relative height of the two modes can get lost. By sampling over both modes we make sure, that no information gets lost.

9.At the end of Sec. 5.1, the authors comment on the difference between the marginalised and profile likelihood saying that the two methods ask different questions. I suggest that the authors elaborate more on that and make clearer which questions are asked and answered.

-> We removed the statement in the text to avoid confusion for the reader. Overall, the difference between both methods is that by profiling we address the most likely value of the Wilson coefficient, while marginalization asks for the most probable point. These are two different statistical frameworks that require two different evaluation methods of the Markov chains.

Attachment:

diff.pdf

---

## Round 1 · Referee Report · Anonymous (Referee 2) · 2023-8-20

Strengths

1- Comprehensive 2- Well written

Report

The manuscript addresses the SMEFT analysis of the EW sector, aimed for setting limit to coefficients on exp data. Specially, the manuscript compares different treatment of nuisance coeffient, whether to profile their likelihoods, or integrate them out.

In section 2 introduces the SM effective field theory Lagrangian, include order 6 terms.

Section 3 details the Bayesian setup, termed SFitter framework. This section It elaborates on the marginal likelihood function of the nuisance parameter, integration over one parameter for a singular measurement, then combination of channels, and finally validate some 1d and 2d distribution.

Section 4 compares the profile and marginal approach for EW sector based on run 2 results.

Section 5 carries out a global SFitter analysis. After validating the marginalization technique in SFitter and presenting a set of new observables, the analysis of the Higgs and electroweak sector after Run 2 is provided. It emphasizes a comparison of the results with a profile likelihood treatment and elucidates the disparities between the two methods using the same dataset. The manuscript points out that the updated dataset reveals inconsistencies between marginalization and profiling treatments of a particular likelihood, stemming from an under-fluctuation in the kinematic mWW distribution.

The study showcased in this manuscript is comprehensive and of superior quality, pertinent to data processing at the LHC. The document is articulately written, with clear presentation of assumptions and details. Queries posed at the outset are effectively addressed later in the text.

Requested changes

A few minor suggestion:

1- Introduce the terms better. SMEFT, SFitter, RFit, ... may not be familiar with readers.
2- After equation 4, "we will not be sensitive to the sign of the muon Yukawa", what will be sensitive, rewording the sentence? 3- Figure 1 may worth commenting more on the difference between the distribution between two approaches 4- Figure 23, may worth comment on the change on distribution shape. 5- in section 6 may worth addressing some outlook at the LHC.

  • validity: top
  • significance: top
  • originality: high
  • clarity: top
  • formatting: perfect
  • grammar: perfect

Author:  Nina Elmer  on 2023-12-22  [id 4203]

(in reply to Report 2 on 2023-08-20)

We thank the referee for the careful revision of our manuscript. Here, we address the various points raised in their reports and we explain how the manuscript has been modified.

  1. Introduce the terms better. SMEFT, SFitter, RFit, ... may not be familiar with readers.

-> We added an explanation of the abbreviation SMEFT, the terms SFitter and RFit are describing analysis tools where we linked the corresponding citations.
 

  1. After equation 4, "we will not be sensitive to the sign of the muon Yukawa", what will be sensitive, rewording the sentence?

-> The problem is that the decay process H-> mumu needs an interference term in order to be sensitive to the sign of the muon Yukawa. But there are no known interference effects.

  1. Figure 1 may worth commenting more on the difference between the distribution between two approaches

->sigma_{flat,tot} indicates in both panels the total uncertainty. For the right panel 3 flat uncertainties were used while in the left panel only one which is why the total uncertainty defined as the quadratic sum of the 'half-widths' is not the same.  We added this explicitly to the caption of Fig. 1.
 

  1. Figure 23, may worth comment on the change on distribution shape. 

-> The results in Fig. 23 show the analysis once including the high kinematic distribution from the WW search and once without it. The main point we want to address in this Figure is, that the underfluctuations in the tail region of the WW search are responsible for the appearing two-mode structure as seen in the top row. Furthermore it also has an impact on the constraining power of f_W, f_3W and f_phid1 since it narrows the peak compared to the results without including the WW search.
 

  1. In section 6 may worth addressing some outlook at the LHC.

-> We added a sentence mentioning the High Lumi analysis in the Appendix.

Attachment:

diff2.pdf

---

## Round 1 · Referee Report · Anonymous (Referee 3) · 2023-8-23

Strengths

1-First very careful exploration of the impact of treatment of correlations and uncertainties in global EFT fits.
2-Very clear explanations of the profiling and marginalisation methods used.

Report

The paper is well written and definitely appropriate for publication. It is discussing a very important topic for global EFT fits, i.e. the difference between profiling and marginalising. This is the first time someone explores this in such a clear and comprehensive way.

I only have a few questions/comments for clarification, mainly with regards to the modelling of the signal and whether the Monte Carlo uncertainties the authors show in some of the plots are reasonable.

Requested changes

1-Even though it is not critical for the purpose of this study it is maybe worth justifying why the authors are employing this basis of operators rather than the most commonly used Warsaw one.
2-It might be worth adding a couple of lines clarifying why the Profiled likelihoods are not smooth in the plots whilst the marginalised ones are.
3-In some of the plots, such as figure 12 and 15, the Monte Carlo error for the Sfitter signal seems to be very large. If this is indeed so large, what was the obstacle for improving that? Order one MC error (and bigger) seems too big to me.
4-I am not entirely convinced why the authors do not compute the qg channel for boosted Higgs at one-loop level. The tools exist to do that.

  • validity: high
  • significance: high
  • originality: high
  • clarity: high
  • formatting: excellent
  • grammar: excellent

Author:  Nina Elmer  on 2023-12-22  [id 4205]

(in reply to Report 3 on 2023-08-23)

We thank the referee for the careful revision of our manuscript. We address the various
points raised in the report and explain the modifications made in the manuscript.

1.Even though it is not critical for the purpose of this study it is maybe worth justifying why the authors are employing this basis of operators rather than the most commonly used Warsaw one.

-> Because we are following the physics arguments of Ref. (32, arxiv: 1211.4580), a translation between the two bases is provided in the Appendix of Ref(67, arxiv: 2108.01094).

2.It might be worth adding a couple of lines clarifying why the Profiled likelihoods are not smooth in the plots whilst the marginalised ones are.

-> The reason for this effect is that marginalization smoothes out statistical fluctuations, which the profile likelihood does not
. We also added an additional comment on page 10 regarding that issue.
 
3.In some of the plots, such as figure 12 and 15, the Monte Carlo error for the SFitter signal seems to be very large. If this is indeed so large, what was the obstacle for improving that? Order one MC error (and bigger) seems too big to me.

-> The errors in figure 12 and 15 are both from the MadGraph event generation and thus Monte Carlo uncertainties. Since both are high kinematic distributions MadGraph had problems populating the tail regions. Also for the considered backgrounds we had to take more than one possible process into account and then add all the individual uncertainties to one for the total background distribution. These factors are responsible for the large MC errors in Figure 12 and Figure 15. In order to improve and reduce the sampling and error one could have used samplings with a higher invariant mass and combine these results with the current ones. However there is a problem doing this, since the MC results are also used for the SMEFT predictions which are later used in SFitter for the global analysis. This could lead to wrong results on the Wilson coefficient constraints.
 

4.I am not entirely convinced why the authors do not compute the qg channel for boosted Higgs at one-loop level. The tools exist to do that.

-> Our implementation is the most economic description in the given pt range. A study validating this point can be found in Ref. (91, arxiv: 1410.5806).

Attachment:

diff3.pdf

Author:  Nina Elmer  on 2023-12-22  [id 4204]

(in reply to Report 3 on 2023-08-23)

We thank the referee for the careful revision of our manuscript. We address the various
points raised in the report and explain the modifications made in the manuscript.

1.Even though it is not critical for the purpose of this study it is maybe worth justifying why the authors are employing this basis of operators rather than the most commonly used Warsaw one.

-> Because we are following the physics arguments of Ref. (32, arxiv: 1211.4580), a translation between the two bases is provided in the Appendix of Ref(67, arxiv: 2108.01094).

2.It might be worth adding a couple of lines clarifying why the Profiled likelihoods are not smooth in the plots whilst the marginalised ones are.

-> The reason for this effect is that marginalization smoothes out statistical fluctuations, which the profile likelihood does not
. We also added an additional comment on page 10 regarding that issue.
 
3.In some of the plots, such as figure 12 and 15, the Monte Carlo error for the SFitter signal seems to be very large. If this is indeed so large, what was the obstacle for improving that? Order one MC error (and bigger) seems too big to me.

-> The errors in figure 12 and 15 are both from the MadGraph event generation and thus Monte Carlo uncertainties. Since both are high kinematic distributions MadGraph had problems populating the tail regions. Also for the considered backgrounds we had to take more than one possible process into account and then add all the individual uncertainties to one for the total background distribution. These factors are responsible for the large MC errors in Figure 12 and Figure 15. In order to improve and reduce the sampling and error one could have used samplings with a higher invariant mass and combine these results with the current ones. However there is a problem doing this, since the MC results are also used for the SMEFT predictions which are later used in SFitter for the global analysis. This could lead to wrong results on the Wilson coefficient constraints.
 

4.I am not entirely convinced why the authors do not compute the qg channel for boosted Higgs at one-loop level. The tools exist to do that.

-> Our implementation is the most economic description in the given pt range. A study validating this point can be found in Ref. (91, arxiv: 1410.5806).

---

## Editorial Decision

resubmitted